# Effects of the sample matrix on the photobleaching and photodegradation of toluene-derived secondary organic aerosol compounds

Alexandra L. Klodt[1], Marley Adamek[2], Monica Dibley[2], Sergey A. Nizkorodov[1*], Rachel E. O'Brien[2*]

[1]Department of Chemistry, University of California Irvine, Irvine, CA, 92697, USA
[2]Department of Chemistry, William & Mary, Williamsburg, VA, 23187, USA

*Correspondence to*: Rachel E. O'Brien (reobrien@umich.edu) or Sergey A. Nizkorodov (nizkorod@uci.edu)

**Abstract.** Secondary organic aerosol (SOA) generated from the photooxidation of aromatic compounds in the presence of oxides of nitrogen ($NO_x$) is known to efficiently absorb ultraviolet and visible radiation. With exposure to sunlight, the photodegradation of chromophoric compounds in the SOA causes this type of SOA to slowly photobleach. These photodegradation reactions may occur in cloud droplets, which are characterized by low concentrations of solutes, or in aerosol particles, which can have highly viscous organic phases and aqueous phases with high concentrations of inorganic salts. To investigate the effects of the surrounding matrix on the rates and mechanisms of photodegradation of SOA compounds, SOA was prepared in a smog chamber by photooxidation of toluene in the presence of $NO_x$. The collected SOA was photolyzed for up to 24 h using near-UV radiation (300-400 nm) from a Xenon arc lamp under different conditions: directly on the filter, dissolved in pure water, and dissolved in 1 M ammonium sulfate. The SOA mass absorption coefficient was measured as a function of irradiation time to determine photobleaching rates. Electrospray ionization high resolution mass spectrometry coupled to liquid chromatography separation was used to observe changes in SOA composition resulting from the irradiation. The rate of decrease in SOA mass absorption coefficient due to photobleaching was the fastest in water, with the presence of 1 M ammonium sulfate modestly slowing down the photobleaching. By contrast, photobleaching directly on the filter was much slower. The high-resolution mass spectrometry analysis revealed an efficient photodegradation of nitrophenol compounds on the filter but not in the aqueous phases, with relatively little change observed in the composition of the SOA irradiated in water or 1 M ammonium sulfate despite faster photobleaching than in the on-filter samples. This suggests that photodegradation of nitrophenols contributes much more significantly to photobleaching in the organic phase than in the aqueous phase. We conclude that the SOA absorption coefficient lifetime with respect to photobleaching and lifetimes of individual chromophores in SOA with respect to photodegradation will depend strongly on the sample matrix in which SOA compounds are exposed to sunlight.

## 1 Introduction

Brown carbon (BrC) aerosol has important impacts on the Earth's radiative forcing (Feng et al., 2013). Because it absorbs actinic radiation in the near-UV (300 to 400 nm) and visible spectral ranges, BrC aerosol reduces the scattering of solar radiation and cooling effect relative to non-absorbing aerosols (Laskin et al., 2015). BrC aerosol can be produced from biomass burning or various gas-phase or multiphase reactions in the atmosphere (Laskin et al., 2015). The photooxidation of aromatic compounds in the presence of nitrogen oxides ($NO_x$) is a known anthropogenic source of BrC (Lee et al., 2014; Liu et al., 2016; Romonosky et al., 2016). For example, secondary organic aerosol (SOA) formed from toluene is brown in appearance because it contains nitrophenols and other chromophoric species (Jang and Kamens, 2001; Lin et al., 2015).

After its formation, the composition and optical properties of BrC continue to slowly change, driven in part by direct and indirect photolysis processes (collectively referred to as "photodegradation" in this paper) (Hems et al., 2021). Studies of BrC aging by exposure to actinic radiation have largely focused on photodegradation processes occurring in cloud-water (Hems et al., 2021), although aerosol particles are estimated to spend about 85% of their lifecycle under non-cloud conditions, i.e. RH <100% (Pruppacher and Jaenicke, 1995). Less work has been done on photodegradation of BrC in submicron particles, which can include

pockets of concentrated aqueous solutions as well as solid organic and inorganic phases with a limited amount of water in them. Water in deliquesced aerosol particles differs from cloud water in several ways, especially in terms of the concentrations of inorganic ions. While the dominant inorganic species in both deliquesced aerosol particles and cloud water are generally ammonium and sulfate (Bikkina et al., 2017), aqueous particles have ionic strengths of greater than 1 M as compared to $10^{-5}$ to $10^{-2}$ M in cloud water (Herrmann et al., 2015). Photochemical processes occurring in the water present in deliquesced aerosol particles

may potentially be altered by high concentrations of inorganic species.

    Previous work has found complex effects of ionic strength on aqueous photochemistry in solutions meant to mimic deliquesced aerosol particles. These studies have largely looked at changes in the UV and visible absorption spectrum and photodegradation rate of single molecules in the presence of inert salts. For instance, the absorption spectrum of pyruvic acid experiences a bathochromic shift of the major absorption band along with the hyperchromic effect at increased ionic strength at lower (< 4) pH

values (Mekic et al., 2018; Luo et al., 2020). However, the photodegradation rate has been shown to increase when the ionic strength was increased with $NaClO_4$ (Mekic et al., 2018), but decrease when the ionic strength was adjusted with NaCl and $CaCl_2$ (Luo et al., 2020). When similar experiments were conducted for lignin-derived compounds, increasing ionic strength lead to formation of a new major absorption band at longer, more atmospherically relevant, wavelengths (Zhou et al., 2019; Loisel et al., 2021). Similarly to pyruvic acid, photodegradation kinetics were observed to accelerate for acetosyringone in the presence of

$NaClO_4$ (Zhou et al., 2019), but decelerate for vanillin in the presence of $NaNO_3$ and $Na_2SO_4$ (Loisel et al., 2021). To the best of our knowledge, only one study has considered the impact of varying ionic strength on the rate of photochemical reactions of complex mixtures of organics representative of aerosol particles. Ray et al. (2020) irradiated solutions of rice-straw smoldering primary organic aerosol in the presence of NaCl, $NaNO_3$, and $Na_2SO_4$ to explore anionic effects and ionic strength effects on photo-bleaching kinetics. They found longer lifetimes of absorbing species in the presence of all ionic species studied. From these studies,

it seems that the specific ionic species present during photolysis may be important; however, no photodegradation studies have been conducted in the presence of ammonium sulfate at high ionic strengths although ammonium and sulfate are the dominant contributors to ionic strength in deliquesced aerosol particles (Bikkina et al., 2017).

    Direct and indirect photolysis have been identified as potentially important sinks for SOA compounds in the condensed-phase environment found in dry organic particles (Romonosky et al., 2016), but there have only been a few studies of these processes.

This environment is uniquely different from aqueous solutions in that SOA molecules are in a highly viscous organic matrix, and different photochemical mechanisms operate compared to those acting in water. Early indications that dry SOA is easily degraded by near-UV radiation – both UV-A and UV-B – came from examining production of volatile products of SOA irradiated directly on its collection substrate (Walser et al., 2007; Mang et al., 2008; Pan et al., 2009). Subsequent studies suggested that direct photolysis is an important sink for atmospheric SOA, but focused primarily on low-$NO_x$ terpene ozonolysis SOA and found that

peroxide and carbonyl compounds were driving the photochemistry (Henry and Donahue, 2012; Hung et al., 2013; Epstein et al., 2014; Wong et al., 2015; Hodzic et al., 2015; Badali et al., 2015; Krapf et al., 2016). As a counter example, Kourtchev et al. (2015) found limited changes in molecular composition of SOA by photolysis compared to heterogeneous oxidation by hydroxyl (OH) radicals for SOA suspended in an environmental simulation chamber, but it should be recognized that separating gas-phase and condensed-phase photochemistry in chamber experiments is challenging. Romonosky et al. (2016) measured the absorption

coefficients of a number of SOA types and estimated their photodegradation lifetimes, concluding that strongly light-absorbing
SOA were predicted to have the shortest lifetimes. Subsequent experiments found that a stronger absorption coefficient does not
necessarily translate to faster photodegradation, with SOA from aromatic precursors being more photostable than SOA from
terpenes, even though the latter barely absorb near-UV radiation (Malecha and Nizkorodov, 2016; Malecha et al., 2018; Baboomian
et al., 2020). While these studies all gave insight into the mechanism of SOA photodegradation during UV-irradiation, none of
them directly measured photobleaching rates in the organic phase.

This study was prompted by a hypothesis that the mechanism of SOA photodegradation in particles dominated by organic material
(e.g., dry SOA with low inorganic content) should be different from that dominated by liquid water (e.g., cloud droplets containing
dissolved SOA). This hypothesis was tested by UV-irradiation of toluene high-$NO_x$ SOA directly on filters and in aqueous solution
and then contrasting the reaction products and absorption spectra. Our second goal was to explore the importance of inorganic salts
in photobleaching and photodegradation of aqueous SOA, and this was done by UV-irradiation of high-$NO_x$ toluene SOA in a
concentrated ammonium sulfate solution and pure water. We present high-resolution mass spectrometry, offline-AMS, and FTIR
results to provide an analysis of changes in chemical composition during photodegradation in different conditions, as well as UV-
Visible spectroscopy results to monitor decay of the overall absorbance by SOA. We find that the compositional changes differ
significantly between the organic particle and aqueous photodegradation. The presence of 1 M aqueous ammonium sulfate does
not appear to significantly impact the changes in SOA composition, but the photobleaching kinetics are modestly slowed as
compared to the experiment in pure water.

## 2 Methods

Experiments were performed at the University of California Irvine (UCI) and at the College of William and Mary (WM). A
summary of the types of the experiments performed and datasets collected is provided in Table S1. Briefly, all SOA filter samples
were prepared in the UCI smog chamber. Some filter samples were mailed to WM on dry ice for on-filter photodegradation
experiments and FTIR analysis. All aqueous phase photodegradation experiments, photobleaching kinetics measurements, and
aerosol mass spectrometry experiments were carried out at UCI.

### 2.1 Secondary Organic Aerosol Generation

SOA was prepared in a ~5 m$^3$ Teflon chamber described previously (Malecha and Nizkorodov, 2017). The chamber was first
humidified to 40% RH. Then $H_2O_2$ was injected through a heated inlet to achieve a mixing ratio of 2 ppm, followed by an injection
of 1.5 ppm of toluene and 0.7 ppm of NO. UV-B lamps within the chamber with an emission spectrum centered at 310 nm were
used to initiate photooxidation, with typical OH steady-state concentrations of about $6\times10^6$ molecule cm$^{-3}$, similar to our previous
work (Hinks et al., 2018). OH steady-state concentrations in the chamber were determined from the rate of toluene depletion using
a Proton Transfer Time of Flight Mass Spectrometer (PTR-ToF-MS; Ionicon model 8000, Innsbruck, Austria) as described in the
SI (see Fig. S1). For the PTR-ToF-MS analysis, the drift tube operated at 60.0 °C ($T_{drift}$), 2.30 mbar ($P_{drift}$), and 600 V ($U_{drift}$). Mass
calibration for the PTR-ToF-MS was performed using three ions commonly observed in room air: $m/z$ 21.0226 ($H_3^{18}O^+$), 29.9980
($NO^+$), and 59.0497 ($C_3H_6O^+$). No seed particles were used in the chamber experiments. A Scanning Mobility Particle Sizer (TSI
Model 3936), $NO_y$ monitor (Thermo Scientific Model 42i-Y), and ozone monitor (Thermo Scientific Model 49i) were used during
all experiments to track particle size distribution and NO, $NO_y$, and ozone mixing ratios. An Aerosol Mass Spectrometer (AMS)
was also used in some experiments to observe composition during aerosol generation and collection. Particles were collected onto
polytetrafluoroethylene (PTFE) filters (Millipore 0.2 μM pore size), then the filters were sealed and frozen at -20 °C until the aging

experiments were performed. Data related to the chamber experiments, including SMPS, NO$_y$, and online-AMS datasets are available online at the Index of Chamber Atmospheric Research in the United States (ICARUS) (Klodt, 2022). Table 1 provides a summary of the finalized set of samples used in this work (it does not include additional samples prepared for preliminary tests to optimize the experimental conditions).

**Table 1:** Summary of samples prepared and the experiments they were used for at William and Mary (WM) and UC Irvine (UCI).

| Filter Number (location of further experiments) | Max SOA concentration in the chamber ($\mu$g m$^{-3}$) | Total SOA collected (mg) | Data sets collected[a] |
|---|---|---|---|
| 1 (WM) | 308 | 0.9 | Online AMS, FTIR |
| 2 (WM) | 328 | 1.1 | Online AMS, FTIR |
| 3 (UCI) | 225 | 1.0 | H$_2$O & Filter Offline AMS, ESI(+,-) |
| 4 (UCI) | 159 | 0.6 | AS & H$_2$O kinetics, ESI(+,-) |
| 5 (UCI) | 178 | 0.8 | AS & H$_2$O kinetics[b] |
| 6 (UCI) | 208 | 0.8 | AS & H$_2$O kinetics[b] |
| 7 (UCI) | 126 | 0.8 | Filter kinetics[b] |
| 8 (UCI) | 196 | 0.6 | Filter kinetics[b] |
| 9 (UCI) | 156 | 0.7 | Filter kinetics[b] |

Footnote: [a]2 filters for FTIR experiments, 1 filter for splitting between aqueous and filter aging, 3 filters for splitting between aqueous and ammonium sulfate aging kinetics, and 3 filters for only filter aging kinetics. [b]AS & H$_2$O kinetics refers to kinetic experiments in 1 M ammonium sulfate and pure water solutions, while filter kinetics refers to kinetic experiments on the filter.

**2.2 Photolysis Experiments**

SOA samples were exposed to near-UV radiation from a Xenon arc lamp (Newport Model 66902). The broadband light was reflected at a 90$^0$ angle with a dichroic mirror, and then passed through a 295 nm long-pass filter (Schott WG295) and a UV bandpass filter (Schott BG1) to remove UVC and visible wavelengths, leaving the majority of the radiation between 280 and 400 nm. A comparison of our lamp spectrum and ambient sunlight is shown in Fig. S2 and Table S2. The comparison shows that, in terms of the 280-400 nm radiation dose, 1 h in our photolysis setup is equivalent to approximately 1.7 h under the 24-hour average Los Angeles solar flux (average taken for June 20$^{th}$) as calculated using the Quick TUV calculator (ACOM: Quick TUV, 2019). Photolysis was carried out for 5 h through the side of a 0.5 cm quartz cuvette containing SOA solution exposed to open air for the aqueous samples or with a filter quarter placed such that the filter surface was uncovered and open to laboratory air in the same position as the cuvettes used in the aqueous experiments. Three photolysis experiments were performed for each experimental condition, as well as one additional experiment for photolysis in pure water and photolysis on-filter to be used for offline-AMS and UPLC-PDA-HRMS, as described in Table 1. Further experimental details on the two setups are provided in the following sections.

**2.2.1 Aqueous Photolysis Experiments**

Aqueous UV-irradiation of the samples was carried out at UCI. After several preliminary experiments to determine the best combination of experimental parameters, including extraction methods and aging conditions, three filters of SOA (numbered 4, 5, 6 in Table 1) were collected. Each filter was cut in half so that half of the filter could be extracted with acetonitrile and then

photolyzed in water and the other half extracted with acetonitrile and then photolyzed in 1 M ammonium sulfate. Each filter was weighed whole, and then a half of the filter was weighed for a more accurate estimation of the mass of aerosol used in each experiment. The SOA material was extracted from the filter half by shaking it gently for 10 min in 5 mL of acetonitrile. The acetonitrile was then removed by rotary evaporation at room temperature (to avoid losing more volatile SOA compounds), and 2.1 to 3 mL of milliQ water (18.2 MΩ-cm, generated using a Thermo Scientific Barnstead NANOpure system) or 1 M ammonium sulfate solution was added to the SOA residue for the photolysis experiments. The SOA dissolved readily in the solution, so no further shaking was necessary. The SOA mass concentration in the solutions was close to 250 mg L$^{-1}$ in all trials (the actual mass concentrations for individual trails are shown in Table S3), and the unadjusted pH of the solutions was about 4.5 in both the ammonium sulfate and pure water conditions. These values are comparable to atmospheric cloud water, which generally contains about 200 mg L$^{-1}$ of dissolved organics and has pH values of between 3 and 6, but the concentrations of organics used here are likely lower than found in particulate matter (Collett et al., 2002, 2008; Herrmann et al., 2015). The solutions were split into three 0.7 to 1 mL (depending on the mass of SOA on the filter) aliquots to serve as the unaged, dark aged, and photolyzed samples. The aliquot designated as "unaged" was prepared immediately for mass spectrometry analysis as described in the next paragraph.

As high concentrations of inorganic ions should not be used with electrospray mass spectrometry, it was necessary to remove the ammonium sulfate from the SOA solutions before performing mass spectrometry analysis. Salt removal for mass spectrometry analysis is commonly accomplished using solid phase extraction (SPE) (Dittmar et al., 2008). However, SPE for inorganic salt extraction has been shown to reduce the recovery of more highly oxygenated and polar compounds as compared to organic solvent extracts, and these compounds are important for our analysis of SOA composition (Kourtchev et al., 2020). Therefore, we chose to extract with organic solvent rather than use SPE. To this end, the majority of the water was removed by rotary evaporation at room temperature. Water removal was stopped when the ammonium sulfate began to precipitate out of solution, but care was taken to not evaporate to dryness, as evaporation of water in the presence of ammonium sulfate has been shown to cause condensation-type reactions that change the SOA composition (Nguyen et al., 2012). The SOA was extracted from the vial containing ammonium sulfate by adding 5 mL of acetonitrile in two 2.5 mL rinses (the addition of acetonitrile caused most of the remaining ammonium sulfate to precipitate and all liquid was removed from the vial). This acetonitrile was also removed by rotary evaporation at room temperature, and a 1:1 mixture of water:acetonitrile was added to the SOA such that the SOA concentration was about 350 mg L$^{-1}$, assuming complete extraction, for mass spectrometry analysis. The same procedure was used for the solutions which did not contain ammonium sulfate for the sake of consistency.

**2.2.2 On-Filter Photolysis Experiments**

Preliminary on-filter photolysis experiments, from Filters 1 and 2 in Table 1, were performed at WM (experimental details are described in the SI section) and then replicated at UCI to allow better intercomparison with the aqueous results using the same photolysis set-up. Both photobleaching kinetics and UPLC-PDA-HRMS experiments for the on-filter samples were performed at UCI. Filters 7, 8, and 9 were cut into quarters, and each quarter was weighed to get a better estimate of the mass of toluene SOA used for each data point. For aging, one filter quarter was placed in front of the photolysis setup such that the filter surface was uncovered and open to laboratory air. While not being photolyzed, the other filter quarters were re-sealed and kept frozen. After photolysis, the filter quarter was extracted with acetonitrile to take a UV-Vis spectrum with the instrument described above. Since UV-Vis sample preparation was destructive, each time point required an entire filter quarter and so only four time points could be taken per filter. Based on the observed photobleaching rate, 0, 1, 3, and 5 h were chosen as time points.

## 2.3 Sample and Data Analyses

### 2.3.1 UV-Vis Analyses

A UV-Vis spectrometer (Shimadzu UV-2450) was used to observe the change in optical properties during photolysis in water or 1 M ammonium sulfate solution. The spectrometer was a double-beam instrument, and the sample solvent was used in the reference cell for each experiment (water for samples collected in water, 1 M ammonium sulfate for samples collected in 1 M ammonium sulfate, etc.). The UV-Vis spectra for the aqueous samples were collected in the 0.5 cm quartz cuvettes used for aging. Data collection involved moving the whole cuvette from in front of the irradiation set up into the UV-Vis spectrometer, taking the spectrum, and then returning the sample to the irradiation set up. Therefore, each UV-Vis spectrum was taken only once per sample at each time point. To keep the organic concentration similar for the on-filter and aqueous photolysis samples, the same 0.5 cm cuvettes were also used after extraction for the samples photolyzed on the filter, although the UV-Vis spectra were taken in acetonitrile (rather than water) after it was confirmed that the spectra did not look different between the two solvents (see Figs. S3 and S4). In the samples where a baseline shift was observed at long wavelengths, corrections were applied by averaging the absorbance from 680 to 700 nm and subtracting the average from all wavelengths (there are no strong absorbers in the SOA at these wavelengths). These corrections were never greater than 0.011 absorbance units. At the sample concentrations used here, the UV-Vis absorbance was less than 1 at all wavelengths longer than 250 nm and within the linear range of the instrument. The UV-Vis data were used to calculate the mass absorption coefficient (MAC):

$$MAC(\lambda) = \frac{A_{10}(\lambda) \times \ln(10)}{b \times C_{org}} \tag{1}$$

where $A_{10}(\lambda)$ is base-10 absorbance, $b$ is the path length, and $C_{org}$ is the mass concentration of the SOA sample in solution (g cm$^{-3}$). Additionally, SOA recovery for each step of the procedure was roughly determined by UV-Vis spectroscopy as the experiment went on, and the results of these checks are shown in Fig. S3 and accompanying description. Comparison of the post-extraction spectrum with the initial spectrum show that the extraction procedure generally recovered about 50-70% of the SOA from the ammonium sulfate solution and greater than 90% if ammonium sulfate was not present. The reason for the retention of some 30-50% SOA by the wet ammonium sulfate residue is unclear and will be investigated in the future.

### 2.3.2 UPLC-HRMS Analysis

Changes in the molecular composition of the UV-irradiated SOA were analyzed by high-resolution mass spectrometry (HRMS) similar to as described previously (Chin et al., 2021). Briefly, the instrument was a Thermo Q-Exactive Plus mass spectrometer (Thermo Scientific) with a resolving power of $1.4 \times 10^5$ at $m/z$ 400 equipped with a heated electrospray ionization inlet. The instrument was operated in both positive (spray voltage +3.5 kV) and negative ion modes (spray voltage -2.5 kV). Ultrahigh performance liquid chromatography (UPLC) and photodiode array (PDA) detection (scanning 190 to 680 nm) were performed to analyze the relative contributions of individual compounds to the total SOA absorption. The column was Phenomenex Luna Omega Polar C18, 150 × 2.1 mm, with 1.6 μm particles and 100 Å pores. The UPLC solvent gradient was 95% solvent A (water acidified to pH 3 with 0.1% formic acid) and 5% solvent B (acetonitrile acidified with 0.1% formic acid) for minutes 0 to 3, followed by a linear ramp to 95% solvent B and 5% solvent A from 3 to 14 minutes, a hold at 95% solvent B from 14 to 16 minutes, and a linear ramp back to 95% solvent A and 5% solvent B for 16 to 22 minutes in preparation for the next run. Separating the samples via liquid chromatography provided the benefit of reducing matrix effects and preventing ionization suppression from any inorganic ions not removed during the extraction process. The separation additionally allowed us to assign formulas to specific peaks in the

PDA chromatogram and therefore better quantify formulas that decreased or remained stable in abundance during photolysis, improving characterization.

Analysis of the PDA-HRMS data was performed using FreeStyle 1.6 from Thermo Scientific, and the peaks with the greatest absorbance when the PDA chromatogram was integrated from 300 and 700 nm were correlated to the peaks in the total ion chromatogram (TIC) based on the instrument's PDA-MS time delay of 0.06 min. Molecular formulas for these chromophores were determined using FreeStyle. Additionally, FreeStyle was used to integrate over the full total ion chromatogram and generate a raw time-integrated (1 to 18 min) mass spectrum. Decon2LS (https://omics.pnl.gov/software/decontools-decon2ls) was used to extract peak positions and relative intensities from the time-integrated mass spectrum, and peaks representing $^{13}$C compounds were removed. Peaks from the blank, unaged, and two aging conditions were aligned with a tolerance of 0.0005 $m/z$. A blank sample was prepared by repeating the SOA extraction and mass spectrometry preparation process with a clean filter. Peaks that were present in the blank at the same or greater intensity as the samples were also removed. Finally, the mass spectra were assigned assuming an accuracy of 0.0005 $m/z$ with a formula of $[C_cH_hO_xN_{0-3}S_{0-1} + Na]^+$ and $[C_cH_hO_xN_{0-3}S_{0-1} + H]^+$ for positive ion mode and $[C_cH_hO_xN_{0-3}S_{0-1} - H]^-$ for negative ion mode (although no sulfur-containing compounds were identified with greater than 0.01% abundance of the maximum peak height). Assigned peaks were used to verify the internal calibration of the $m/z$ axis in both ion modes and adjust the calibration if needed. The internal calibration improved the $m/z$ accuracy and usually led to a few additional assignments for peaks that could not be assigned within 0.0005 $m/z$ in the uncalibrated mass spectra. Finally, neutral formulas were determined from the assignments, and the data from the positive and negative ion modes were clustered together. The mass spectra presented below show the combined peak abundance in the positive and negative ion mode data referenced to formulas of the unionized SOA compounds.

### 2.3.3 FTIR Analysis

A Shimadzu IR Tracer-100 MIRacle 10 with a diamond crystal ATR probe was used to collect ATR-FTIR spectra from 600-4000 cm$^{-1}$. A total of 45 scans were averaged per sample, and for each sample, an air background was collected before the filter was adhered to the crystal. After irradiating the filter segments for 0 (control), 6, 18 and 24 h, small slivers of the filters were cut off and pressed onto an ATR-FTIR crystal (diamond) using the swivel press. The filters were then detached, an unpressed area of the filter was moved overtop the crystal and the filter was pressed onto the crystal again. After the second press, the filters were removed from the crystal, leaving behind a thin film of sample material. The spectra of the adhered SOA without the Teflon filter were collected. The spectra were converted to absorbance and the baseline for each spectrum was corrected using a Baseline Spline Fit (http://wavemetrics.com/project/BaselineSpline) using an Akima spline. The baseline corrected spectra were normalized to the total absorbance and smoothed with a three-point rolling average to improve inter-comparison.

### 2.3.4 Offline AMS Analysis

A High-Resolution Time-of-Flight Aerosol Mass Spectrometer (HR-ToF-AMS or AMS; Aerodyne, Billerica, MA, USA) operated in V-mode and a custom ultrasonic small volume nebulizer, described elsewhere (O'Brien et al., 2019), were used to analyze the composition during and after photolysis. For AMS analysis, particles were vaporized at 600 °C and ionized using electron impact ionization at 70 eV. For the ultrasonic small volume nebulizer, sample preparation involved combining ~3 μL of sample solutions with ~2 μL of internal standard solution, consisting of a mixture of 0.25 g L$^{-1}$ isotopically labelled NH$_4$$^{15}$NO$_3$ and 0.25 g L$^{-1}$ NH$_4$I. The 5 μL of prepared sample were then loaded onto a clean Kapton film in the nebulizer. The sample was nebulized, and the aerosol particles were carried with a flow of clean air into the inlet of the AMS. The AMS data were analyzed with Igor Pro

(version 7.0.8.1, WaveMetrics Inc.) using the ToF-AMS Analysis Toolkit 1.63H and ToF-AMS HR Analysis 1.23H software packages. The signals for the $NO^+$ and $NO_2^+$ ions were quantified for each injection and intercompared between different samples after calculating the ratios with the signals for the corresponding isotopically labeled ions from the ammonium nitrate internal standard ($^{15}NO^+$ and $^{15}NO_2^+$).

## 3 Results and Discussion

### 3.1 Optical Properties

Figure 1(a) shows how the wavelength-dependent MAC of toluene SOA changes with photolysis in water. MAC data for the other photolysis and dark conditions are provided in Fig. S4. The decrease in absorption between 300 and 700 nm over the five hours of photolysis was used to calculate the photobleaching lifetime of the chromophoric compounds in the SOA. Previous work on toluene SOA suggests the absorption band at 350 nm, the most prominent peak in the spectrum, is mostly attributable to the $\pi$ to $\pi^*$ transition of nitrophenols, such as methylnitrocatechol (Lin et al., 2015).

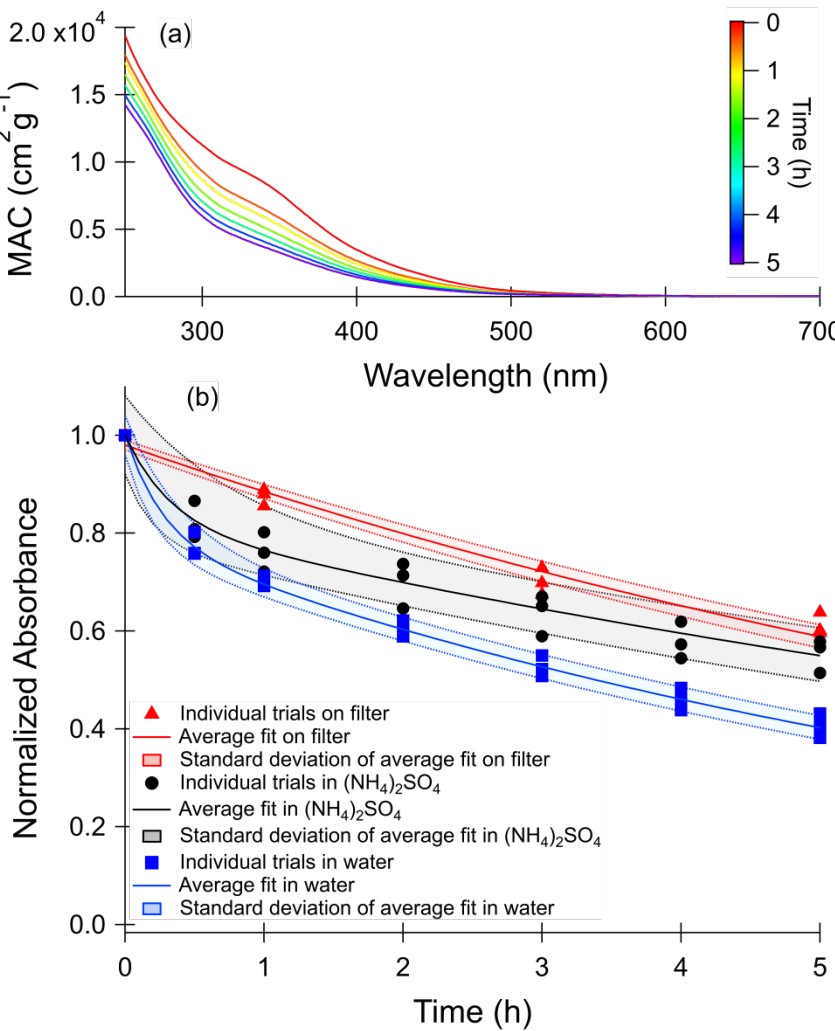

**Figure 1:** (a) Wavelength-dependent mass absorption coefficient (MAC) values recorded during photolysis of toluene SOA in water. MAC plots for other conditions, shown in Fig. S4, are qualitatively similar. (b) Normalized absorbance decay of the wavelengths integrated from 300 to 700 nm with photolysis. Photolysis experiments in 1 M ammonium sulfate are shown in black circles, in pure water are shown in blue squares, and on the filter are shown in red triangles. Values on the y-axis were corrected

for changes in the dark and then normalized to the mass absorption coefficient at zero minutes. Shaded areas represent one standard

deviation of the fit of the three combined trials.

The time-dependent change in absorbance integrated from 300 to 700 nm for all photolysis conditions is shown in Fig. 1(b). In the case of the aqueous samples, a correction was applied for absorbance changes occurring in the dark by subtracting the change in integrated MAC as compared to time zero for the dark condition from the integrated MAC of the corresponding photolysis

condition. This was particularly necessary in the 1 M ammonium sulfate trials because a small peak slowly forms at 300 nm under these conditions in the dark (see Fig. S4) as a result of ammonia-carbonyl reactions.(Laskin et al., 2015) The integrated MAC values for the dark controls and a fit to the MAC changes are shown in Fig. S5. The absorbances over time were then normalized to the initial absorbance (t=0) and were fit to the biexponential decay shown in equation (2),

$$y(t) = A_1 e^{-k_1 t} + A_2 e^{-k_2 t} \tag{2}$$

where $y(t)$ is the normalized absorbance as a function of time, $k_1$ and $k_2$ are first-order rate constants for a faster and a slower process, and $A_1$ and $A_2$ are relative normalized absorbance contributions of compounds decaying through the faster and slower processes, respectively. We stress that this is a purely empirical fit, as in reality there are a large number of light-absorbing compounds in SOA, each with its own complex time dependence.

Based on this analysis, we observed a considerably slower decay in MAC on-filter than in the aqueous phase. However, for the

filter sample we could only collect 4 data points, which may introduce a bias in our analysis. In particular, the on-filter data did not include the 0.5 h time point, which characterizes the fast-reacting chromophore pool in the aqueous samples. As a result, $A_1$ was constrained to zero for the filter samples to avoid over-fitting, and this fit is shown in Fig. 1(b). The fit for the on-filter data can be improved slightly by including a parameter representing a photorecalcitrant fraction ($R^2$=0.99 compared to $R^2$=0.97), shown in Fig. S6. Previous studies have found significant fractions of photorecalcitrant material when SOA photolysis was performed in

the organic particle phase, including for toluene high-NO$_x$ SOA (O'Brien and Kroll, 2019; Walhout et al., 2019; Baboomian et al., 2020; Pospisilova et al., 2021), and based on these studies we expect a photorecalcitrant fraction for the SOA studied here. When the photorecalcitrant fraction was not constrained to zero, the fit obtained estimates a large photorecalcitrant absorbance fraction (49 ± 5%). This fraction of photorecalcitrant absorbance is similar to the Baboomian et al. (2020) estimate of 50% photorecalcitrant mass fraction after photolysis of toluene-derived high-NO$_x$ SOA deposited on a gold surface. Future longer-term photolysis

experiments on filters are needed quantify what fraction of the absorbance is fully photorecalcitrant in the atmosphere versus much slower than the rates observed for the aqueous samples.

Calculated fitting parameters from Fig. 1(b) are summarized in Table S4. Absorbance lifetimes calculated from these values and scaled to the 24-h average solar actinic flux in Los Angeles are shown in Table 2. The shortest photobleaching lifetimes are seen for photolysis in water, while the lifetimes are about twice as long in 1 M ammonium sulfate.


**Table 2:** Absorbance lifetimes for photobleaching processes described by Eq (2) scaled to the 24-h average solar actinic flux in Los Angeles. The unscaled measured rate constants are provided in Table S4. Errors represent the standard deviations over the three combined trials.

|  | $\tau_1$ (h) | $\tau_2$ (h) |
|---|---|---|
| **H$_2$O** | 0.5 ± 0.12 | 12.6 ± 0.7 |
| **1 M ammonium sulfate** | 0.5 ± 0.30 | 21 ± 2.7 |
| **Filter** | NA | 17 ± 1.0 |

**3.2 Chemical Composition Changes with Photolysis**

**3.2.1 PDA Data**

In an effort to tie the photobleaching behavior observed in the UV-Vis data to changes in composition, UPLC-PDA-HRMS analysis was performed on the samples before and after aging. PDA data for the photolysis conditions are shown in Fig. 2 – unaged, photolyzed in water, and photolyzed on filter from Filter 3 (Fig. 2a) and unaged, photolyzed in water, and photolyzed in 1 M ammonium sulfate from Filter 4 (Fig. 2b). The PDA counts were integrated from 300 to 680 nm to correspond to the analysis of
the UV-Vis data. Under all conditions, a large fraction of the eluting compounds was unresolved, forming a broad peak stretching from 5 to 12 min. Superimposed on top of the unresolved peak were several well-resolved peaks. Additionally, some loss of resolution is seen between the two panels in Fig. 2. We attribute this to deterioration of the HPLC column as the data in Fig. 2b were taken nearly a year before the data in Fig. 2a. To keep our conclusions robust, we compared the changes between the unaged and photolyzed in water samples, which are included in both experiments, and have excluded peaks which were not reproducible
between the two trials from our discussion. In the unaged chromatogram, the most abundant species corresponding to major PDA peaks are all nitrophenol-type compounds. Those to which we were able to assign a chemical name based on previous work (Lin et al., 2015) include: $C_7H_7NO_3$ (two structural isomers of nitrocresol 10.06 and 10.36 min), $C_7H_7NO_4$ (three structural isomers of methylnitrocatechol - 8.04, 9.14, and 9.69 min), $C_6H_5NO_3$ (nitrophenol - 9.35 min), and $C_6H_5NO_4$ (nitrocatechol – 8.37). The peaks for which the name is not known are marked with an asterisk in Fig. 2. The major peaks marked in Fig. 2 are also the most abundant
compounds in the integrated mass spectrum shown in Figs. 3(a) and 4(a).

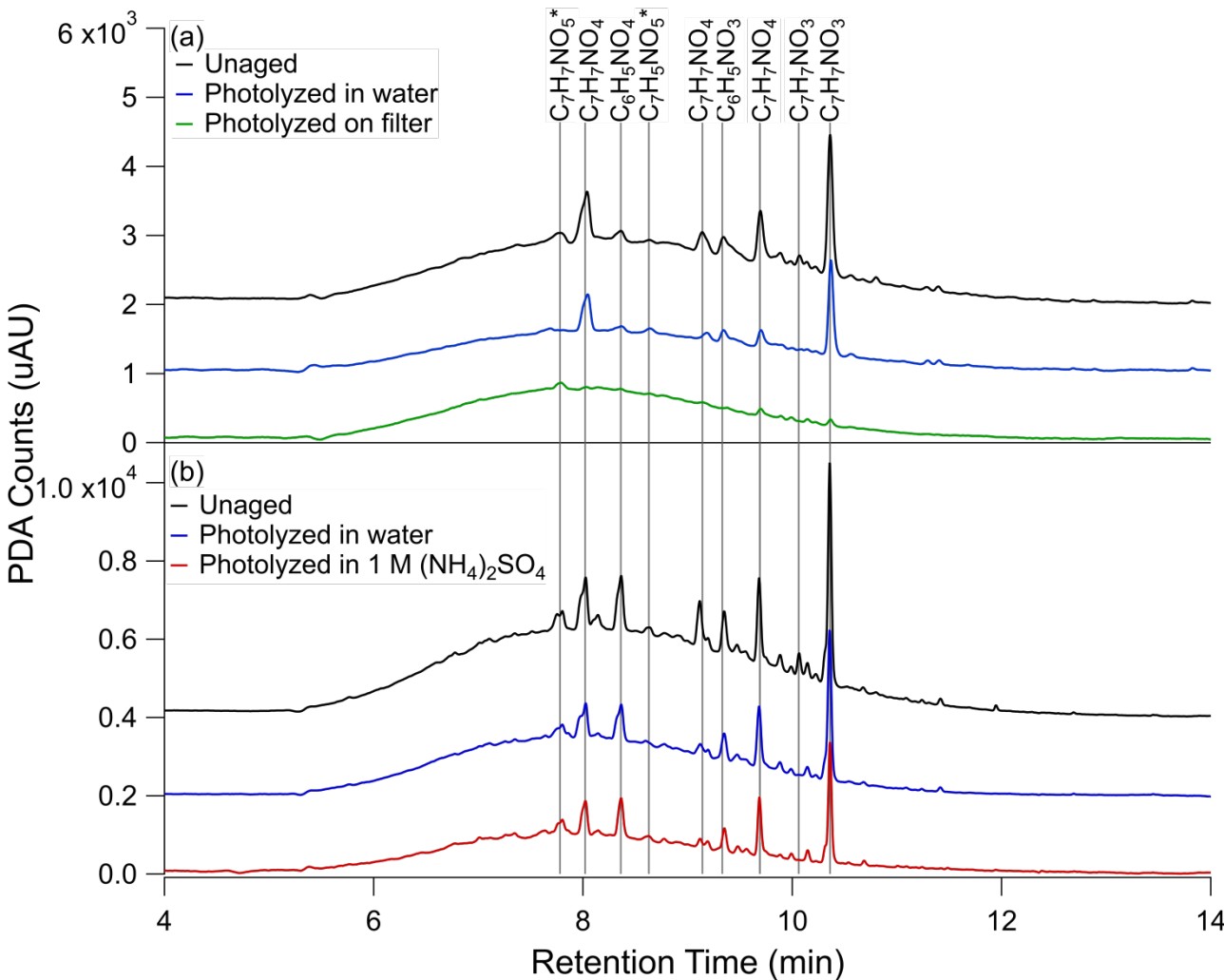

**Figure 2:** UPLC-PDA chromatograms for (a) unaged toluene SOA sample (black trace), photolyzed 5 h in water (blue trace), and photolyzed 5 h on the filter (green trace) from Filter 3 and (b) unaged toluene SOA (black trace), photolyzed 5 h in water (blue trace), and photolyzed 5 h in 1 M ammonium sulfate (red trace) from Filter 4. The black (unaged) and blue (photolyzed in water) traces are shown twice to illustrate the extent of reproducibility of this analysis. PDA counts were integrated over 300 to 680 nm wavelength range to match with the UV-Vis data analysis. A blank PDA spectrum, also integrated from 300 to 680 nm, was subtracted from each PDA spectrum shown here. The blank PDA chromatogram can be seen in Fig. S7, along with the PDA data for the control samples. The peaks for which molecular formulas could be assigned are marked on the graph and their assigned formulas are indicated, while asterisks denote unnamed formulas. In the case of 1 M ammonium sulfate, PDA counts were adjusted for the estimated 50% extraction efficiency. Additionally, the baselines are each offset by $10^3$ PDA counts on the y-axis of panel (a) and $2 \times 10^3$ PDA counts on the y-axis of panel (b) for ease of comparison.

In Fig. 2(a), a significant difference can be seen in the PDA data after photolysis on the filter compared to photolysis in water. These differences are further quantified in Table S5. After 5 h of photolysis in water (blue trace) there is only about a 40% change in the resolved features in both trials shown in Fig. 2, but after 5 h of photolysis on the filter (green trace) the resolved features are greatly reduced – by about 90%. By contrast, the area of the unresolved absorbance in the filter photolysis decreased by about 20% compared to the unaged sample (black trace) while the unresolved absorbance in the aqueous photolysis trace decreased by 35%.

The control conditions (Fig. S7) show little change in resolved or unresolved peak area with dark aging, generally within 10% of the unaged condition.

Figure 2(b) compares the PDA absorbance with photolysis in water and 1 M ammonium sulfate. The addition of ammonium sulfate produced results similar to pure water in the PDA chromatogram, although the lower extraction efficiency may influence these results. For instance, our calculations in Table S5 suggest the baseline feature decreased more with photolysis in 1 M ammonium sulfate (45%) than in water (35%) but considering the similar decrease in resolved peak area in the two conditions (both about 40%) and the slower photobleaching observed in the UV-Vis data with 1 M ammonium sulfate, this is likely an effect of extraction

efficiency. If some compounds preferentially remain with the ammonium sulfate after extraction, we will miss them in this analysis.

### 3.2.2 SOA Composition – High Resolution Mass Spectrometry

Unaged toluene SOA composition has been well characterized using HRMS previously (Lin et al., 2015), so we will focus our discussion on composition changes before and after photolysis. Figure 3 compares the retention time-integrated SOA mass spectra for photolysis in water and on-filter (from Filter 3), while Fig. 4 contrasts the behavior of SOA from a separate experiment in water

and in 1 M ammonium sulfate (from Filter 4). In all cases, the mass spectra represent combined positive and negative ion modes scaled to the approximate mass concentration of organics in the samples. The mass spectra, including the sample extracted from 1 M ammonium sulfate, show excellent reproducibility in terms of peak height of all major peaks and the shape of peak distribution after accounting for the concentration of organics in the mass spectrometry samples (also see Fig. S8 and Fig. S9), demonstrating that the extraction method used here is effective for extracting SOA samples for mass spectrometry analysis. The unaged SOA

mass spectra are dominated by nitrogen-containing compounds, the most abundant of which are $C_7H_7NO_3$ (153 Da – nitrocresol) and $C_7H_7NO_4$ (169 Da – methylnitrocatechol), in agreement with previous work on high-NOx toluene SOA (Jang and Kamens, 2001; Lin et al., 2015; Hinks et al., 2018). It should be noted that these are also the most abundant peaks observed in the PDA data in Fig. 2.

A visual comparison of the highest peak intensities in the mass spectra in Fig. 3(a) (unaged) and 3(b) (photolyzed in water) suggests

that photolysis in water does not lead to a large change in the overall composition. There is some decrease in the abundance of nitrogen-containing compounds upon UV-irradiation, but many of the major peaks are the same as in the unaged spectrum. Some formation of CHO compounds at lower molecular weights (less than 150 Da) can be seen. These are not observed in the dark-aged aqueous samples (see Fig. S8 and Fig. S9). The peaks that are most different in abundance between the photolysis and dark samples (such that they are among the 10 most abundant peaks after photolysis but not after dark aging) are present at much greater

abundances in the negative mode than the positive mode, and can be assigned to small organic acids, such as maleic acid at 116 Da. Organic acids are established aqueous photolysis products of nitrophenols (Alif et al., 1991).

The difference in the composition after on-filter photolysis (Fig. 3(c)) is much more apparent. In contrast to the aqueous photolysis condition (Fig. 3(b)) where the nitrogen-containing peaks are relatively unaffected, the nitrogen-containing peaks in the organic-rich phase are greatly reduced in abundance with photolysis – such that the CHON compounds no longer dominate the mass

spectrum. In contrast to the large reduction in CHON compounds, the CHO compound abundance appears relatively unchanged. We do not observe an increase in peak abundances at low molecular weights as we did with aqueous photolysis. This suggests the photolysis products may be different in the organic phase as compared to the aqueous phase. We should note that the aqueous photolysis products may remain dissolved in water after formation, while those from filter photolysis may more easily escape into the gas phase if they are formed on the surface of the SOA film.

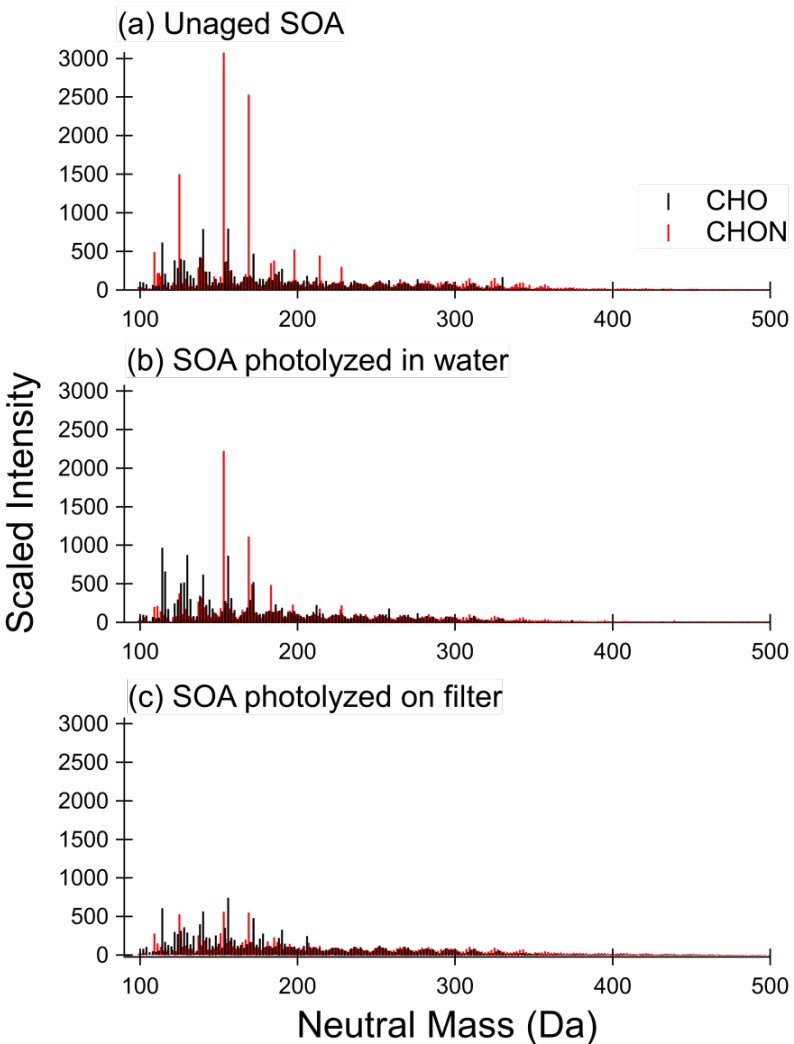

**Figure 3:** Mass spectra for (a) unaged SOA sample, (b) SOA sample photolyzed 5 h in water, and (c) SOA sample photolyzed 5 h on the filter from Filter 3. CHON compounds are shown in red and CHO compounds are shown in black. The signal is scaled to approximate pre-photolysis mass concentration of SOA in mass spectrometry samples. Control (dark aged) samples are shown in Fig. S8.

The unaged samples and the results of aging in water appear reproducible when comparing the two separate experiments depicted in Fig. 3 and 4 – little change is observed other than a modest increase in abundance of low molecular weight CHO compounds with photolysis in water. Further, the addition of 1 M ammonium sulfate (Fig. 4(c)) did not have a significant impact on the composition after photolysis as compared to pure water (Fig. 4(b)). There does appear to be less increase in low molecular weight CHO compounds, which is reasonable considering the photobleaching was about half as fast in the ammonium sulfate condition as the pure water condition. We conclude that ammonium sulfate did not have a large effect on changes in SOA composition with photolysis as compared to pure water, but rather simply slowed down the photolysis rate.

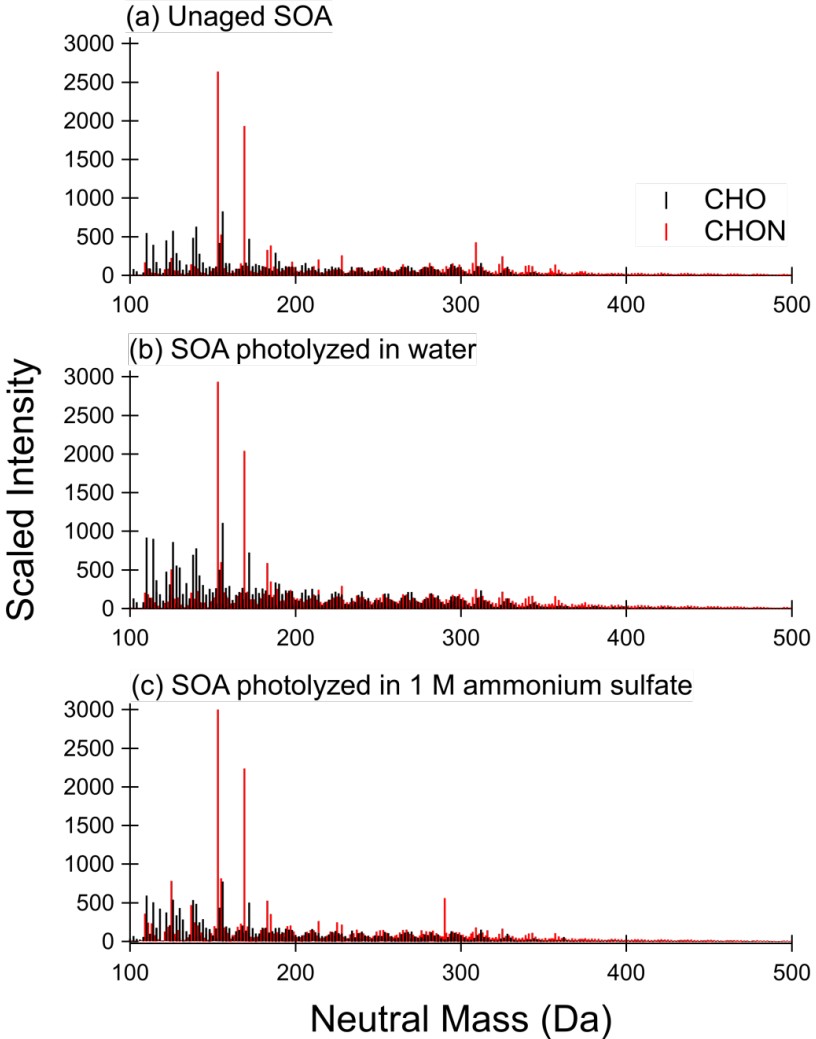

**Figure 4.** Mass spectra for (a) unaged SOA sample, (b) SOA sample photolyzed 5 h in water, and (c) SOA sample photolyzed 5 h in 1 M ammonium sulfate from Filter 4. CHON compounds are shown in red and CHO compounds are shown in black. Signal is scaled to approximate pre-photolysis mass concentration of SOA in mass spectrometry samples. Control (dark aged) samples are shown in Fig. S9.

Visualization methods such as the Van Krevelen diagram are useful in interpreting the large amount of data obtained from high resolution mass spectrometry (Kim et al., 2003; Nozière et al., 2015; Merder et al., 2020). Van Krevelen diagrams for the unaged and photolysis conditions are shown in Fig. 5. For these plots, compounds were sorted into groups of CHO (in black) and CHON (in red), and then H:C and O:C ratios were binned and summed so that each marker on the diagram represents the sum of all compounds with that H:C and O:C ratio, and marker size was scaled to this value. Additionally, only points with intensities greater than one percent of the maximum intensity in the unaged sample are shown in Fig. 5 to simplify this figure. Van Krevelen diagrams containing all measured H:C and O:C ratios are provided in Figs. S10 and S11. The discussion of the Van Krevelen diagrams presented here describes visually identified qualitative changes between conditions. We leave the quantitative discussions to the other techniques presented in this work. In the unaged SOA (Fig. 5(a)), the markers are clustered around an O:C ratio of 0.5 and an H:C ratio of 1.0, which is typical of oxidized aromatic hydrocarbons (Wozniak et al., 2008; Nozière et al., 2015). The largest markers in the unaged samples represent CHON compounds, with the most abundant summed CHO ratios being much less abundant. After photolysis in water in Fig. 5(b), there was a modest reduction in the total abundance of compounds with CHON

formulas and an increase in total abundance of compounds with CHO formulas. The ratios that increased in summed intensity generally had both higher H:C and O:C ratios than the ratios that decreased in summed abundance. There is little change observed after aging in the dark as seen in Fig. S10. The shift in summed ratio abundance observed with photolysis in water was also observed for the sample aged in 1 M ammonium sulfate (Fig.5(c)), although to a lesser extent. It should be noted that the unaged sample shown in Fig. 5(a) is from Filter 3, while the sample photolyzed in 1 M ammonium sulfate shown in Fig. 5(c) was from Filter 4, and the starting summed abundancies of the H:C and O:C ratios were slightly different between the two samples. The Van Krevelen diagrams for the unaged and control samples from Filter 4 are included in Fig. S11, where the shift in summed peak abundance with photolysis in 1 M ammonium sulfate is more evident. Finally, after photolysis on the filter (Fig. 5(d)), there was a dramatic reduction in the summed abundances of CHON compounds. Further, an increase in summed abundances of CHO compounds is not evident, demonstrating that changes in molecular composition with photolysis in water is very different from changes in molecular composition with photolysis on the filter.

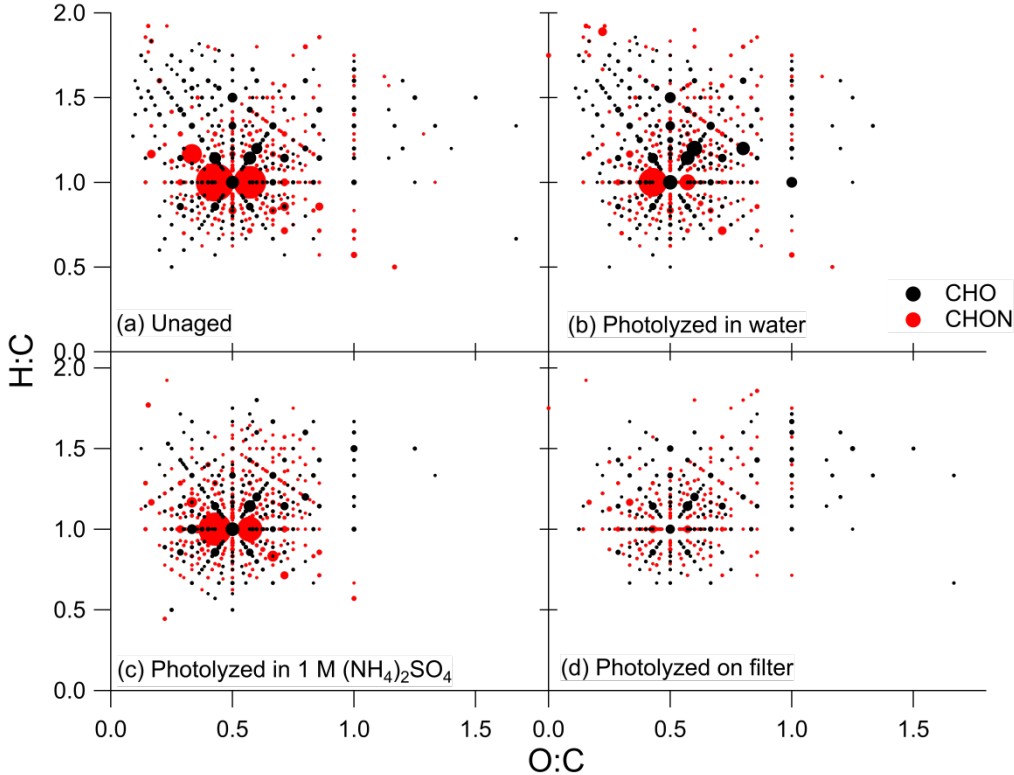

**Figure 5.** Van Krevelen diagrams for the (a) unaged sample from Filter 3, (b) sample photolyzed in water from Filter 3, (c) sample photolyzed in 1 M ammonium sulfate from Filter 4, and (d) sample photolyzed on the filter from Filter 3. CHO formulas are shown in black and CHON formulas are shown in red. The size of the marker represents the summed mass spectrometry abundance of all CHO or CHON compounds with the corresponding H:C and O:C ratios, and all points with intensities less than one percent of the maximum intensity in the unaged sample are excluded. Control (dark aged) Van Krevelen diagrams and all points excluded here are shown in Figs. S10 and S11.

### 3.2.3 Changes in Nitrogen-Containing Groups with Photolysis on Filters

To confirm the removal of nitrogen-containing compounds during on-filter photolysis, nitrogen content was quantified with offline-AMS data. The purpose of this analysis was to directly contrast the difference in composition changes with photolysis. Since we did not observe dramatic changes in chemical composition during the aging of toluene SOA between the water and 1 M ammonium

sulfate conditions, and given the low concentrations of SOA mass on our sample filters, the 1 M ammonium sulfate conditions were excluded from this analysis. This allowed us to perform this analysis on a single filter split between the water and filter aging

conditions, making the SOA composition of the two samples as identical as possible. Analysis was performed by normalizing the $NO^+$ and $NO_2^+$ signal to the labelled $NO^+$ and $NO_2^+$ signals from the internal standard as shown in Fig. 6. The normalization accounts for variations in the signal for each nebulization pulse, which has been demonstrated to occur with the small volume nebulizer (O'Brien et al., 2019). Organonitrates and organic nitro groups generate $NO^+$ and $NO_2^+$ when ionized in the AMS, thus differences in the total amount of these ions suggests a loss of these functional groups in the samples (Farmer et al., 2010). Before

photolysis, the concentrations of nitrogen-containing compounds for the aqueous and on-filter samples are slightly outside of each other's standard deviations, which may be a result of small differences in the amount of labeled ammonium nitrate added to the solutions or a small variation in extraction efficiencies between the two filter halves. There is no apparent change in nitrogen content after 5 h of photolysis in pure water. In stark contrast, there is a reduction in nitrogen content after 5 h of photolysis directly on the filter. Inorganic nitrate groups may also contribute to the $NO^+$ and $NO_2^+$ signals. However, organic and inorganic nitrate

fragment with different $NO^+/NO_2^+$ fragment ratios (Fry et al., 2009; Bruns et al., 2010), with fragment ratios from ammonium nitrate being lower than fragment ratios from nitrogen-containing organic compounds. We calculated the $NO^+/NO_2^+$ fragment ratios for our samples as well as for the isotopically labeled ammonium nitrate standard that was added to the samples, and these results are shown in Fig. 6(b) and 6(c). After photolysis, the $NO^+/NO_2^+$ fragment ratios in both the aqueous and filter conditions decreased somewhat (Fig. 6(b)), but not to the level of the $NO^+/NO_2^+$ fragment ratios from the inorganic standard (Fig. 6(c)). We

therefore do not believe the resistance of the aqueous sample to losing nitrogen-containing peaks with photolysis is solely due to interference with inorganic nitrate, although the results here do suggest some change in the relative amounts of organic and possibly inorganic nitrogen-containing groups with photolysis. FTIR data, shown in Fig. S12, also fully support the loss of nitrogen-containing groups with on-filter photolysis. Peaks corresponding to both organonitrates and nitroaromatics decreased over time with photolysis relative to carbonyl peaks (more details are provided in the SI). Both the AMS and the FTIR results agree well

with the relatively qualitative ESI results.

The removal of nitrogen-containing compounds from the on-filter samples suggests that the nitrogen-containing groups present in the SOA are being converted to gas-phase products that volatilize from the filter. Indeed, photolysis of gaseous ortho-nitrophenols was shown to produce HONO (although there was significant disagreement on the HONO quantum yield).(Bejan et al., 2006; Sangwan and Zhu, 2016) Photolysis of para-nitrophenols in solutions, including viscous solutions containing organic solutes, has

also been shown to produce $HONO/NO_2^-$.(Barsotti et al., 2017)  It is likely that $HONO/NO_2^-$ production is suppressed in a condensed-phase environment by excitation quenching. The extent of this suppression could be stronger under aqueous conditions compared to an organic matrix, resulting in slower removal of nitrogen from the photolyzed aqueous sample. In addition, HONO would more easily volatilize from the filter than from an aqueous solution further reducing the AMS signal from nitrogen compounds in on-filter photolysis experiments.

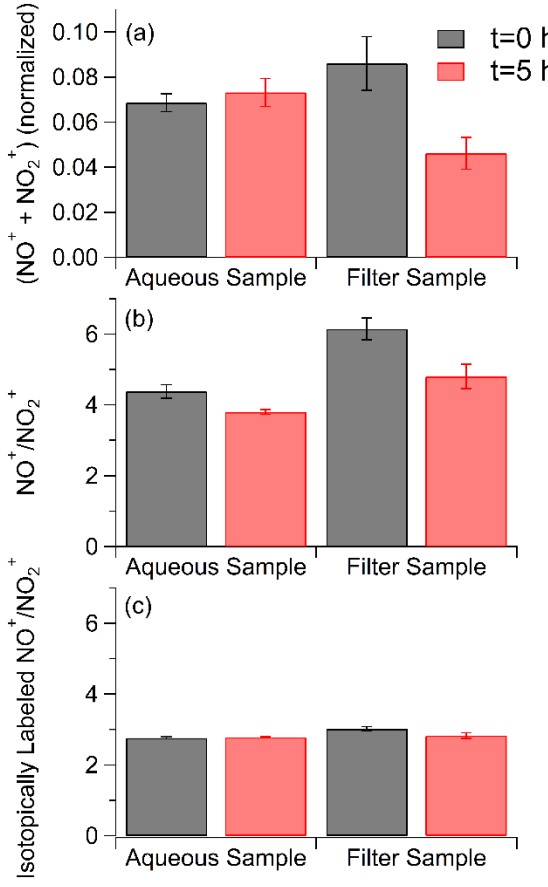


**Figure 6**. (a) Totals for the main fragment ions for nitrogen-containing groups measured by aerosol mass spectrometry before (in black) and after 5 h of photolysis (in red) in the aqueous phase and organic particle phase. $NO^+$ and $NO_2^+$ fragment concentrations are normalized to an internal standard. (b) $NO^+/NO_2^+$ fragment ratios before (in black) and after 5 h of photolysis (in red) for samples age in water and on the filter. (c) $NO^+/NO_2^+$ fragment ratios for the isotopically labeled inorganic internal standard. Error

bars represent one standard deviation over five injections.

### 3.2.4 Effect of Photolysis Matrix on Photo-degradation Mechanism

It is clear from the UPLC-PDA-HRMS data that the photolysis of toluene SOA proceeds differently in the viscous organic phase compared to the aqueous phase, and here we aim to propose plausible explanations for our observed experimental results based on previous studies. Our experiments show that nitrophenols, which are mostly resolved in the PDA data and are expected to be major

chromophores in this SOA, are preferentially photolyzed in the viscous organic phase. Previous studies have established that nitrophenols photolyze more quickly in the organic phase than in the aqueous phase, and the proposed mechanism is abstraction of a hydrogen atom from surrounding organic molecules by a triplet state of an excited nitrophenol (Lignell et al., 2014; Dalton and Nizkorodov, 2021). For 3-nitropyrene, hydrogen abstraction was observed to be nearly diffusion controlled, while charge transfer reactions were not observed in organic solvents (Scheerer and Henglein, 1977). Further, it was observed that phenolic

hydrogens were most efficiently abstracted by the triplet excited states of nitro-polycyclic aromatic hydrocarbons (PAH), with other types of easily-abstractable hydrogens not participating in photodegradation (Feilberg and Nielsen, 2000). We expect a significant fraction of phenolic compounds in our SOA, and in the organic phase an excited triplet state will be in closer proximity to abstractable phenolic hydrogens than in the aqueous phase, making this a plausible explanation for relatively fast removal of

CHON compounds in the organic phase. Additionally, oxy-PAHs with an *n* to $\pi^*$ triplet character have been shown to strongly accelerate nitro-PAH decay through the initiation of a radical chain reaction (Feilberg and Nielsen, 2000). This pathway is more important in the organic phase than in the aqueous phase because differences in matrix polarity change the relative energies of the *n* to $\pi^*$ and $\pi$ to $\pi^*$ transitions for many aromatic carbonyls, such that the more reactive *n* to $\pi^*$ transition is favored in less polar matrices and the less reactive $\pi$ to $\pi^*$ transition is favored in polar matrices (Horspool and Lenci, 2003). These mechanisms in combination likely explain the relatively rapid loss of nitrophenols in the organic phase as observed by mass spectrometry.

In spite of *slower* photodegradation of nitrophenols (assessed via HRMS, offline-AMS, and FTIR analyses), the overall photobleaching (assessed via UV-Vis analyses) was actually *faster* in the aqueous phase as compared to on-filter photodegradation. Since we observed minimal composition change with mass spectrometry, photolysis in the aqueous phase must result in the decay of highly absorbing compounds with low relative abundance or poor electrospray ionization efficiency. This observation correlates well to the observations in Lin et al. (2015), which catalogued the chromophores in toluene SOA, and attributed 40 to 60% of the PDA absorbance to poorly separated chromophores with low peak abundances in ESI mass spectra (Lin et al., 2015). Additionally, this may agree with the larger decrease in the unresolved baseline feature observed in our aqueous PDA samples (-35% and -45% for the pure water and ammonium sulfate conditions, respectively) as compared to our on-filter PDA samples (-21%). The chromophores which are being photolyzed in the aqueous conditions contribute strongly to the sample absorbance, but not necessarily to the mass spectrum. Further, the fact that photobleaching is slowed with the addition of ammonium sulfate suggests that these lower abundance compounds may still react through a triplet state when photolyzed. Ionic species may quench an excited triplet compound before it reacts (Treinin and Hayon, 1976), leading to slower chromophore loss with the addition of ammonium sulfate.

There is also a possible role of reactions between aqueous ammonia and carbonyl compounds in SOA, which are known to cause browning through the formation of imine-type chromophores (Laskin et al., 2015). While these reactions were shown to contribute to only a modest change in the MAC of high-$NO_x$ SOA from aromatic hydrocarbon precursors (Updyke et al., 2012) and the nitrogen-containing products formed through these processes tend to have short photolysis lifetimes (Hems et al., 2021), aqueous photolysis reactions would lead to continuous formation of these small carbonyl compounds (Moran and Zepp, 1997) and potentially continuous formation of these imine-type chromophores. We do see a peak slowly forming at about 300 nm in the dark with ammonium sulfate, which has a similar peak position to that previously observed for trimethylbenzene high-$NO_x$ SOA aged with ammonium (Updyke et al., 2012). Unfortunately, it is hard to directly observe these compounds by our mass spectrometry methods (Bones et al., 2010), because they have been shown to hydrolyze in the mass spectrometry solvent (Laskin et al., 2010). Therefore, we suggest that reaction of carbonyls with dissolved ammonia play only a secondary role in the photochemistry of aqueous toluene SOA in presence of ammonium sulfate.

**4 Conclusions/Atmospheric Implications**

Toluene high-$NO_x$ SOA was aged by direct photolysis on filters simulating the viscous organic phase in aerosol particles and in aqueous solution with and without 1 M ammonium sulfate representing dilute atmospheric cloud water and concentrated aerosol-phase water, respectively. UV-Vis measurements reported here show that photobleaching is fastest in pure water, while the presence of ammonium sulfate modestly slows the rates of photobleaching. Photobleaching in the viscous organic phase is much slower and likely results in the formation of photorecalcitrant compounds. Chromatographic analysis showed that nitrophenols, as well as a large number of unresolved chromophores, contribute to the absorption coefficient of this type of SOA. We find that the decay of nitrophenol compounds proceeds more quickly in the viscous organic phase than in the aqueous phase, in agreement with

previous studies measuring the photolysis of single nitrophenols in aqueous solutions and in viscous organics (Dalton and Nizkorodov, 2021). In contrast, the SOA composition of the aqueous phase samples does not change appreciably over the 5 h photolysis experiment in spite of the observed faster overall photobleaching (assessed by UV-Vis analysis). This suggests preferential photodegradation of unidentified chromophores with high absorptivity but low peak abundance in the mass spectra.

Based on previous studies, we propose that the differences in photolytic aging may be explained by differences in triplet state reactivity between sample matrices. Previous work has observed different photolysis rates for the same chromophore in different atmospherically-relevant environments, such as water, alcohol solutions, and glassy organic matrices (Fleming et al., 2020; Dalton and Nizkorodov, 2021; Hinks et al., 2016; Lignell et al., 2014). These studies mostly focused on the photolysis of select nitrophenol compounds and found nitrophenols to photolyze faster in the organic solvents than in water. Our results confirm that SOA nitrophenols, as a group, appear to the more efficiently photolyzed chromophores in the organic phase. However, this cannot be generalized to all chromophores because we observe that other, presently unidentified, chromophores are more efficiently photolyzed in aqueous solution. Therefore, generally speaking, photolysis rates of individual major chromophores can be quite different from the overall photobleaching rates, particularly when measured in the aqueous phase, and the study of individual chromophores does not fully represent the behavior of complex SOA particles.

Reaction with the OH radical is considered the most important sink for organic molecules in the atmosphere. In the case of SOA, biomass burning organic aerosol, and individual nitrophenol compounds, OH reactions are often observed to cause chromophore formation for a short time period before leading to overall photobleaching for both heterogeneous reactions on particles and reactions in the aqueous phase (Hems et al., 2021). Subsequent photobleaching of various BrC aerosol types have a wide variety of OH lifetimes ranging from hours to days (Hems et al., 2021), which is on the same order we measure for our UV-irradiation experiments. Nitrophenols react relatively quickly with OH in the aqueous phase – lifetimes of several hours have been reported for a few representative compounds – so OH will likely be a more important sink for nitrophenols in clouds and aqueous aerosol (Zhao et al., 2015; Hems and Abbatt, 2018). However, since we observe that photobleaching occurs much more quickly than nitrophenol loss in the aqueous phase, we can expect both OH reaction and photodegradation to be important factors affecting the lifetimes of individual chromophores in the atmosphere.

The results reported here will likely vary depending on ionic strength or particle viscosity. The ammonium sulfate concentration used here – 1 M – is on the low end of ionic strengths found in atmospheric particles (Herrmann et al., 2015). If our proposed explanation is correct, higher salt concentrations will likely quench triplet reactivity more efficiently and so may slow photodegradation further. Additionally, higher ammonium concentrations may also lead to faster formation of imine-based chromophores. Future work is needed to determine the precise relationship between the ionic strength and photobleaching rate for different types of SOA. Furthermore, this study was limited to ammonium sulfate, but other anions, such as halides, are more effective triplet quenchers than sulfate (Jammoul et al., 2009; Gemayel et al., 2021). More importantly, more work on the impact of the matrix viscosity on photodegradation of SOA should be done. Previous studies have found higher viscosity to sometimes slow down and sometimes accelerate photodegradation of individual nitroaromatics (Hinks et al., 2016; Dalton and Nizkorodov, 2021). The conflicting results are likely due to the specific reactivity of nearby molecules during photolysis as well as triplet lifetime of the absorber. Therefore, it will be useful to independently vary the organic matrix viscosity and the types of surrounding species in future studies in order to fully understand the factors controlling photobleaching and photodegradation lifetimes of SOA. Further work is needed to confirm the underlying mechanism causing the differences in photodegradation and photobleaching rates observed here and in previous studies.

## Data availability

Data related to the chamber experiments, including SMPS, NOy, and online-AMS datasets are available online at the Index of Chamber Atmospheric Research in the United States (ICARUS) (Klodt, 2022)

## Author contribution

ALK, SAN, and ROB conceived the study, ALK, MA, MD, and ROB performed the experiments, ALK wrote the original draft, all authors edited and reviewed the final draft.

## Competing Interests

The authors declare that they have no conflict of interest.

## Acknowledgements

The UCI team acknowledges support from the U.S. National Science Foundation grant AGS-1853639, and thanks Natalie R. Smith for helping take UPLC-PDA-HRMS data during the pandemic time. The UPLC-PDA-HRMS instrument used in this work was purchased with the U.S. National Science Foundation grant CHE-1920242. The WM team acknowledges support from the U.S. National Science Foundation grant AGS-2042619 and William & Mary Honors research funding. The WM team also thanks William McNamara for the use of the arc lamp.

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
