# Peer review of "Effects of the sample matrix on the photobleaching and photodegradation of toluene-derived secondary organic aerosol compounds"

_Atmospheric Chemistry and Physics, 2022_

## Referee Comment (RC2)

In this manuscript, Klodt and co-authors investigate the effect of sample matrix on the photobleaching of brown carbon generated in a smog chamber (toluene + $NO_x$) via UV-vis spectroscopy and liquid chromatography coupled to both UV-vis detection and high-resolution mass spectrometry (HRMS).

This study addresses the broader question on how to correctly model photobleaching of brown carbon in the atmosphere, where presence of a broad range of matrix complicates the understanding and modeling of these processes. Previous studies have mostly focused on cloud-water conditions, and the few that focused on aqueous aerosols conditions used individual compounds, not atmospherically relevant organic mixtures (i.e., organic aerosols). Furthermore, most used salts other than ammonium sulfate, some of which of limited environmental relevance. Klodt et al.'s contribution addresses this knowledge gap by investigating the photodegradation process of a lab-generated organic aerosols under cloud-water, aqueous-aerosol, and dry-aerosol conditions, in the presence and absence of ammonium sulfate.

While motivations and quality of this work certainly justify publication in *Atmospheric Chemistry and Physics*, I believe that the paper would greatly benefit from a more *quantitative* analysis of the available data. My largest criticism concerns the analysis of the HRMS data. Based on the information provided in the text, the authors based their conclusions on "naked-eye", qualitative differences in MS spectra between two experimental conditions. While I acknowledge that performing quantitative analyses of HRMS data of environmental mixtures is not straightforward, I still think the authors can obtain useful (semi-qualitative) information from their spectra. In other environmental chemistry fields, typical analyses include changes (also only qualitative; see Laszakovits and MacKay, 2022[1]) in the population of Van Krevelen diagrams and changes in total number of identified formulas, among other things. I advise the authors to check, e.g., Thorsten Dittmar's publications (Uni Oldenbourg) to identify which analyses are most suitable for their dataset (e.g., see Merder et al., 2020[2]).

In addition to HRMS data, I think that the manuscript would benefit of a more quantitative analysis of UPLC-PDA data (Figure 3) and of additional analyses of UV-vis data (Figures 1-2). Specifically, UPLC-PDA data should be discussed in terms of changes (i.e., photolyzed sample – unaged sample) in the number of detectable peaks and in total areas (either the area of the unresolved baseline feature alone, unresolved baseline + all defined peaks, and, in case, defined peaks of interest). Furthermore, based on their method description, the authors should be able to assign a molecular formula to all well-defined peaks in Figure 3 (a few are briefly mentioned in lines 267 – 270). In theory, commercially available compounds could also be quantified (and/or their structure could be confirmed); thus, for these peaks, changes in *concentrations*, not only areas, could be provided. (While this would be a valuable addition, I acknowledge that it will require additional lab work that the authors may or may not wish to undertake.) These additional pieces of information will give a better understanding of which molecular structures/molecules behave differently under various experimental conditions – which may provide further insights into the underlying photodegradation mechanism (see comments below).

Concerning UV-vis data (Figure 1-2), I have two main suggestions. First, the authors should consider analyzing how *all chromophores* change during irradiation, not only the band at 350 nm. While I understand the rationale that they provide (even though Lin et al. gives

absorbance maximum values from 292 to 360 nm, not only 350 nm), the authors are using a complex mixture, not with a pure nitrophenol. To achieve this, they can consider variations in the total area ≥ 300 nm (or ≥ 290 nm) as a function of irradiation time (corrected for variations occurring in the dark). Repeating their calculation at selected wavelengths (in addition to the 350 nm one) is also an option. The results in Table 2 should then be referred to all chromophores, not only to the ones at 350 nm (the authors can also discuss variations in lifetime when considering specific wavelength or all chromophores).

A second suggestion concerns the choice of the fitting equation. While I agree with using a biexponential decay equation (indeed, this is common in the aquatic photochemistry community; see, e.g., Del Vecchio and Blough[3]. The authors may consider highlighting the parallelism between dissolved organic matter and SOA photobleaching in their text), I found the rational for excluding some components of the equation rather arbitrary. Instead, I suggest fitting all datasets with the general equation (eq 2), and then use parameters related to the goodness of the fit (e.g., $R^2$ or others) to justify the use of other fitting equations. I also advise the authors to remove y0 from eq 2 (unless strictly necessary to have a good fit) and limit their analysis to the 5 h irradiation data. Indeed, I do not agree with their justification for setting y0 = 0 for the aqueous solutions samples: by looking at the ammonium sulfate data in Fig S4, at 24h the amount of absorbance left is still half of that at 5h. I thus believe that more datapoints are required to confidently assess the presence of a photorecalcitrant fraction.

Last, the authors should acknowledge that the fit for the on-filter photolysis samples includes only 4 datapoints (notably, there is less data at lower irradiation times, where the fast-decay pool would be visible); this may be a reason why their data had to be fitted to a monoexponential decay function. I also recommend adding data for the dark controls in Figure 2.

My last major criticism concerns the structure of the materials and methods. In its current form, the authors mention some techniques whose data was not discussed in the paper (i.e., online AMS data during collection of the aerosol samples), and do not describe techniques that are later presented in the paper (i.e., FT-IR data). Furthermore, there should be a more explicit description of which blanks were collected and which results they provided (e.g., in current section 2.3.1, did the authors also performed photolysis on a blank filter?) Last, I think that a different order of the various subsections will improve readability. Specifically, I suggest following this order (some additions/changes are also indicated for each subparagraph).

2.1 SOA generation – In addition to describe the aerosol generation experiment, as they already do, the authors should also provide and briefly discuss (in the supplement, not in the main text) data related to $O_3$, $NO_x$, AMS, particle size, as it will help other researchers reproducing their results. Furthermore, the chamber used in their experiment should be described (or a reference should be provided to a published description of their chamber). The *online* AMS analysis (now part of section 2.3.2) should also be in this section (i.e., in the supplement), not as part of the main text (unless the authors wish to discuss AMS data as a part of their results; in this case, a comparison with HRMS and UV-vis data of filters pre-irradiation should also be included). If all this data was already presented and discussed in a publication, they can simply add the reference to the published work. In addition, description of how blank samples are obtained should be added as well.

2.2 Photolysis experiments – Here, the authors should describe the general features of the irradiation setup used for both aqueous and on-filter photolysis (i.e., xenon lamp, now part

of section 1.2.1). In addition to the information already provided, the authors should also add a description of their irradiation experiments – How long where these experiments run for? When did the authors collect aliquots for which analyses? How many replicates were performed?

2.2.1 Aqueous experiments – Here, the authors should describe specific details related to the aqueous-phase experiments, including pre- and post-irradiation sample preparation (now, also in section 1.2.1)

2.2.2 On-filter experiments – Here, the authors should add details specific to irradiation of filters, including post-irradiation sample preparation for further analyses (now part of section 2.3.1)

2.3 Sample and data analyses

2.3.1 UV-vis analyses – Here, there should be a description of the UV-vis spectrometer and data analysis protocol. In the current form, this section requires some changes - in my opinion, there are too few details on actual measurements and too many details on recovery calculation. Missing information on sample collection include the following: how was the UV-vis operated (single beam or double beam)? If in double-beam, which reference was used? Which cuvettes were used (1-cm quartz)? How many replicates were collected for each datapoint? Missing information on data analyses include the following: Did you perform blank subtraction? (If yes, which blank spectrum did you use?) Did you correct for long-wavelength absorption? Did you correct for changes occurring in the dark? (According to Figure S3, there are some substantial variations in UV-vis absorption in the ammonium sulfate kept in the dark.) What is the range of absorbance you detected at your experimental concentrations? (i.e., was within the linear range of the instrument?) Furthermore, the text on estimating SOA recovery from UV-vis analyses should be moved to the supplement.

2.3.2 UPLC-PDA-HRMS analyses – Here, add content of section 2.2.3. Additional data analyses performed (see previous comments) should also be described here.

2.3.3 FT-IR analyses – Please, provide instrument setting, sample analyses, and data analyses details here. This information is currently missing.

2.3.4 Offline AMS analyses – Here the authors can refer to the O'Brien et al. publication describing the system they used.

Below, I provide a list of minor changes that the authors may wish to address in preparing their revised manuscript.

*Abstract and introduction*
**Lines 8-9:** Please, change "oxides of nitrogen" to "oxides of nitrogen ($NO_x$)".
**Line 14**: Please, change "UV-B" to "UV" (based on your spectrum in Figure S1, you have also a substantial amount of UVA during your experiments).
**Line 24:** Please, change "nitrophenol" to "nitrophenols".
**Line 33:** Please, change "$NO_x$" to "nitrogen oxides ($NO_x$)"
**Line 39**: Please, specify what "non-cloud conditions" refer to.
**Line 44:** Please, change "aqueous particles" to "deliquesced aerosol particles".
**Line 45**: Please, change "particle phase water" to "water present in deliquesced aerosol particles".
**Line 62**: Please specify why ammonium sulfate is a more interesting/relevant salt to use compared to others already investigated in the literature.

**Lines 65-66**: The authors may want to point out that also the concentration range of the organics is different, not only the ionic strength; this can have effect on products formation (e.g., oligomers formation is more likely when solutions are more concentrated).

**Line 67**: Please, specify if "near UV" refers to UVA, UVB, or both. Furthermore, in which phase are the SOAs in the discussed studies? (i.e., was the irradiation performed on filters or in the aqueous phase?)

**Line 79**: Please change "This" to "This hypothesis".

*Materials and methods*

**Line 97:** How was hydroxyl radicals steady-state concentrations measured/estimated? This information should be added in the supplement (where additional information related to section 2.1 of the materials and methods are presented).

**Line 99 and 100:** Is "$NO_y$" a typo? Why not "$NO_x$"?

**Table 1:** Please, change "location" to "location of further analyses or experiments" in the heading of the first column (or add a new column named "location of the experiment").

**Lines 120-121:** Did the extracts dissolve easily in water (or ammonium sulfate solution), or did the authors sonicate the filters? (If the authors sonicated, for how long and at which power?). The authors should also clarify which type of water was used to prepare their aqueous extract (i.e., deionized water or milliQ; if MilliQ water, information on the generation system should also be provided).

**Lines 121, 124, 142**: Please, change "µg/mL" to "mg/L". This applies also to instances in the supplement (e.g., Table S2).

**Line 124**: I found it surprising that also the water solution has such a low pH (generally, the pH of deionized water is 5.5 – 6.5 due to dissolved $CO_2$). Can you explain this lower pH? (e.g., is it due to organic acids parts of your aerosol or to inorganic components? Can the AMS data provide useful insights?)

**Lines 133-134**: Please, clarify if the cuvette was irradiated from the top or from the side.

**Line 135:** A typical approach to remove salts and prepare samples for HRMS analysis involve solid phase extraction (SPE). Have the authors tried/considered this procedure as well? Are there literature precedents showing that your procedure is better/more convenient?

**Line 144:** Please, check that "g/mL" is the correct concentration unit.

**Line 166:** Please, provide the brand of the UPLC instrument.

**Lines 171 – 173:** I would add that separating the compounds by LC previous to HRMS allows one also to improve characterization – as you can assign a formula to each peak of the chromatogram.

*Results, discussion, and conclusions*

**Figures 1-2**: I suggest combining Fig. 1 and Fig.2 in a single 2-panel figure, at it will make it easier for the reader to see that Fig. 2 was obtained using data from Fig. 1. Furthermore, I suggest changing the x-axis of Fig. 2 to "time (hours)" (instead of "(min)"), as this is the same unit in the legend of Fig. 1. In the caption related to Fig. 1 data, the authors should specify the conditions under which this data was acquired, instead of referring to "representative conditions" (i.e., say directly that it is was collected during an aqueous-phase experiment in water). In addition, a reference to spectra collected in other conditions (i.e., Figure 3) should be added to the caption.

**Line 264**: Please, provide empirical evidence for the following statement: "[…] the fastest absorbance decrease with photolysis was observed at 350 nm" – from analysis of their UV-vis data, the authors should be able to justify this choice.

**Figure 3**: I have two suggestions here: 1. Remove one panel (I found it confusing that the black and blue traces are shown twice); to increase readability, the authors may consider increasing the offset of each peak and normalize the PDA counts to one given, convenient, signal; 2. Indicate on the graphs which are the peaks for which you could assign a molecular formula (and add the formulas/names).

**Line 295:** Please, explain how you assessed the reproducibility of your analyses.

**Line 313 – 315**: Here, a total organic carbon analysis before and after irradiation could provide some evidence about this point. If the authors did this analysis or have material left to perform this analysis, it would be good addition to their experimental dataset.

**Figures 4 and 5**: As for the two panels in Figure 3, I advise the authors to produce one single figure with all four experimental conditions.

**Line 343 – 344:** Did the authors record FTIR data also for aqueous samples? Can their offline AMS method distinguish between inorganic (e.g., $HNO_3$) and organic nitrogen-containing compounds? If not, inorganic nitrogen containing products may remain in the aqueous phase contributing to $NO^+ + NO_2^+$ counts.

**Figure 6**: Please explain why the ammonium sulfate data was not included in this analysis.

**Section 3.2.4 and lines 393, 414-415**: Please, make clear that the ones discussed are only *proposed* mechanisms based on previously reported reactivity for nitrophenols. The authors did not perform any experiment to test the underlying mechanism. Instead, this point should be highlighted as an area for further research in the final section. I also highlight that running more analyses to confirm the identity of some of the well-resolved peaks (see my comments above) will allow the authors to collect a piece of evidence for the existence of triplet reactivity – if compounds that they identify are the same for which triplet reactivity has been *observed*, not only speculated, in previous work.

**Lines 375 – 379 (and lines 414-415):** Photodegradation of dissolved organic matter generates carbonyls (e.g., Moran and Zepp[4] or Zhu and Kieber[5]). If the same happens during SOA photodegradation, carbonyls may react with $NH_4^+$ to form new nitrogen-containing compounds. Did the authors consider this mechanism while explaining their ammonium sulfate results?

**References**

(1) Laszakovits, J. R.; MacKay, A. A. Data-Based Chemical Class Regions for Van Krevelen Diagrams. *J. Am. Soc. Mass Spectrom.* **2022**, *33* (1), 198–202. https://doi.org/10.1021/jasms.1c00230.

(2) Merder, J.; Freund, J. A.; Feudel, U.; Hansen, C. T.; Hawkes, J. A.; Jacob, B.; Klaproth, K.; Niggemann, J.; Noriega-Ortega, B. E.; Osterholz, H.; Rossel, P. E.; Seidel, M.; Singer, G.; Stubbins, A.; Waska, H.; Dittmar, T. ICBM-OCEAN: Processing Ultrahigh-Resolution Mass Spectrometry Data of Complex Molecular Mixtures. *Anal. Chem.* **2020**, *92* (10), 6832–6838. https://doi.org/10.1021/acs.analchem.9b05659.

(3) Del Vecchio, R.; Blough, N. V. Photobleaching of Chromophoric Dissolved Organic Matter in Natural Waters: Kinetics and Modeling. *Mar. Chem.* **2002**, *78* (4), 231–253. https://doi.org/10.1016/S0304-4203(02)00036-1.

(4) Moran, M. A.; Zepp, R. G. Role of Photoreactions in the Formation of Biologically Labile Compounds from Dissolved Organic Matter. *Limnol. Oceanogr.* **1997**, *42* (6), 1307–1316. https://doi.org/10.4319/lo.1997.42.6.1307.

(5) Zhu, Y.; Kieber, D. J. Wavelength- and Temperature-Dependent Apparent Quantum Yields for Photochemical Production of Carbonyl Compounds in the North Pacific Ocean. *Environ. Sci. Technol.* **2018**, *52* (4), 1929–1939. https://doi.org/10.1021/acs.est.7b05462.

---

## Author Comment (AC1)

**Response to Referee Comment on acp-2022-278**
**Anonymous Referee #1**

The authors investigated aging of secondary organic aerosols (SOA) produced by mixing hydrogen peroxide (H2O2) (2 ppm), toluene (1.5 ppm) and NO (0.7 ppm) upon UV-B light irradiation. The aerosol particles were collected on PTFE filters and then after extraction photolyzed in pure water and in solution containing 1 M of ammonium sulfate.

Changes in optical properties were monitored by UV-VIS spectroscopy and changes in molecular composition of the irradiated SOA were observed by high-resolution mass spectrometry (UPLC-PDA-HR-MS). The SOA composition was monitored by HR-TOF-AMS.

The authors show that photobleaching is faster in pure water representative of cloud water compared to that in presence of 1m of ammonium nitrate representative of aqueous particles. The photobleaching in the viscous organic phase is much slower compared to that in pure water and in the presence of ammonium sulfate. Nitrophenols and other chromophores were responsible for the absorption of SOA.

The experiments were well performed and the obtained results can be of interest for the readers of Atmospherics Chemistry and Physics (ACP). Indeed, understanding the influence of sample matrix on photodegradation and photobleaching processes is very important issue and needed to resolve the complex puzzle of chemical processes occurring in the atmosphere.

Therefore, I recommend publication in ACP.

I am just curious about the slower photobleaching in solution containing 1 M ammonium sulfate. Is it influenced by the ammonium ions (NH4+) or by the sulfate anions (SO42-)?

We thank the reviewer for the assessment and for this question. At this time, we are not in a position to answer the question posed by the reviewer because we only carried out experiments in ammonium sulfate, and we did not experiment with other salts. Since the effect of the ammonium sulfate on photochemistry was relatively minor, it is not clear whether a more detailed study with different salts is warranted. It will be more interesting to examine pH effects in the future as pH varies over a wide range in atmospheric particles.

---

## Author Comment (AC2)

**Response to Referee Comment on acp-2022-278**
**Anonymous Referee #2**

In this manuscript, Klodt and co-authors investigate the effect of sample matrix on the photobleaching of brown carbon generated in a smog chamber (toluene + $NO_X$) via UV-vis spectroscopy and liquid chromatography coupled to both UV-vis detection and high-resolution mass spectrometry (HRMS). This study addresses the broader question on how to correctly model photobleaching of brown carbon in the atmosphere, where presence of a broad range of matrix complicates the understanding and modeling of these processes. Previous studies have mostly focused on cloud-water conditions, and the few that focused on aqueous aerosols conditions used individual compounds, not atmospherically relevant organic mixtures (i.e., organic aerosols). Furthermore, most used salts other than ammonium sulfate, some of which of limited environmental relevance. Klodt et al.'s contribution addresses this knowledge gap by investigating the photodegradation process of a lab-generated organic aerosols under cloud- water, aqueous-aerosol, and dry-aerosol conditions, in the presence and absence of ammonium sulfate.

While motivations and quality of this work certainly justify publication in *Atmospheric Chemistry and Physics*, I believe that the paper would greatly benefit from a more *quantitative* analysis of the available data. My largest criticism concerns the analysis of the HRMS data. Based on the information provided in the text, the authors based their conclusions on "naked-eye", qualitative differences in MS spectra between two experimental conditions. While I acknowledge that performing quantitative analyses of HRMS data of environmental mixtures is not straightforward, I still think the authors can obtain useful (semi-qualitative) information from their spectra. In other environmental chemistry fields, typical analyses include changes (also only qualitative; see Laszakovits and MacKay, 2022[1]) in the population of Van Krevelen diagrams and changes in total number of identified formulas, among other things. I advise the authors to check, e.g., Thorsten Dittmar's publications (Uni Oldenbourg) to identify which analyses are most suitable for their dataset (e.g., see Merder et al., 2020[2]).

*We thank the reviewer for the suggestion. We have prepared Van Krevelen diagrams (Fig. 5, S10, and S11) for our data sets as well as a discussion of the changes in O:C and H:C distributions with photolysis in lines 379 to 404, which is useful for making our point about composition changes with photolysis in water versus on the filter.*

In addition to HRMS data, I think that the manuscript would benefit of a more quantitative analysis of UPLC-PDA data (Figure 3) and of additional analyses of UV-vis data (Figures 1-2). Specifically, UPLC-PDA data should be discussed in terms of changes (i.e., photolyzed sample – unaged sample) in the number of detectable peaks and in total areas (either the area of the unresolved baseline feature alone, unresolved baseline + all defined peaks, and, in case, defined peaks of interest). Furthermore, based on their method description, the authors should be able to assign a molecular formula to all well-defined peaks in Figure 3 (a few are briefly mentioned in lines 267 – 270). In theory, commercially available compounds could also be quantified (and/or their structure could be confirmed); thus, for these peaks, changes in *concentrations*, not only areas, could be provided. (While this would be a valuable addition, I acknowledge that it will require additional lab work that the authors may or may not wish to undertake.) These additional pieces of information will give a better understanding of which molecular structures/molecules behave differently under various experimental conditions – which may provide further insights into the underlying photodegradation mechanism (see

comments below).

These are excellent suggestions. The requested analysis has been made and new or updated figures have been incorporated in the revised version. Additionally, since the reviewer has suggested that we integrate our UV-Vis data from 300 to 700 nm, we have made the same integration with our PDA data. As a result, we have updated and replaced Fig. 2 and S7, along with adding Table S5. We have also updated our discussion of these plots in various lines between 300 and 332.

Concerning UV-vis data (Figure 1-2), I have two main suggestions. First, the authors should consider analyzing how *all chromophores* change during irradiation, not only the band at 350 nm. While I understand the rationale that they provide (even though Lin et al. give absorbance maximum values from 292 to 360 nm, not only 350 nm), the authors are using a complex mixture, not with a pure nitrophenol. To achieve this, they can consider variations in the total area $\geq 300$ nm (or $\geq 290$ nm) as a function of irradiation time (corrected for variations occurring in the dark). Repeating their calculation at selected wavelengths (in addition to the 350 nm one) is also an option. The results in Table 2 should then be referred to all chromophores, not only to the ones at 350 nm (the authors can also discuss variations in lifetime when considering specific wavelength or all chromophores).

A second suggestion concerns the choice of the fitting equation. While I agree with using a biexponential decay equation (indeed, this is common in the aquatic photochemistry community; see, e.g., Del Vecchio and Blough[3]. The authors may consider highlighting the parallelism between dissolved organic matter and SOA photobleaching in their text), I found the rational for excluding some components of the equation rather arbitrary. Instead, I suggest fitting all datasets with the general equation (eq 2), and then use parameters related to the goodness of the fit (e.g., $R^2$ or others) to justify the use of other fitting equations. I also advise the authors to remove y0 from eq 2 (unless strictly necessary to have a good fit) and limit their analysis to the 5 h irradiation data. Indeed, I do not agree with their justification for setting $y0 = 0$ for the aqueous solutions samples: by looking at the ammonium sulfate data in Fig S4, at 24h the amount of absorbance left is still half of that at 5h. I thus believe that more datapoints are required to confidently assess the presence of a photorecalcitrant fraction.

Last, the authors should acknowledge that the fit for the on-filter photolysis samples includes only 4 datapoints (notably, there is less data at lower irradiation times, where the fast-decay pool would be visible); this may be a reason why their data had to be fitted to a monoexponential decay function. I also recommend adding data for the dark controls in Figure 2.

We thank the reviewer for the useful suggestions. We have modified the UV-vis analysis to focus on the data integrated from 300 to 700 nm. We additionally have simplified equation 2 to exclude y0 and re-fit all data to this new equation. The on-filter data was then re-fit to a single exponential fit when the fitting software returned infinity for the standard deviation of the fitting parameters. Since we do expect some photorecalcitrant fraction for the on-filter samples based on previous work and allowing a fraction of non-decaying absorbance improved the $R^2$ value, we included a brief discussion of the possibility of a photorecalcitrant fraction, added Figure S6, and called for future studies to quantify this value. We also added a clarification about the number of data points and resulting uncertainty in fitting at earlier times. We have updated the data in Figure 1(b), Table 2, Table S4, and the discussion of these results in various lines between 250 and 290. We feel that adding the dark controls to Figure 1(b) would make the Figure too busy. We have instead added Figure S5 to depict the dark

controls. We also removed the previous Figure S4 and the SI section on UV-Vis analysis at WM because we are no longer discussing the 24 h photolysis results and Figure S5 because we feel Figure 1(b) and Table S4 provide the same information.

My last major criticism concerns the structure of the materials and methods. In its current form, the authors mention some techniques whose data was not discussed in the paper (i.e., online AMS data during collection of the aerosol samples), and do not describe techniques that are later presented in the paper (i.e., FT-IR data). Furthermore, there should be a more explicit description of which blanks were collected and which results they provided (e.g., in current section 2.3.1, did the authors also performed photolysis on a blank filter?) Last, I think that a different order of the various subsections will improve readability. Specifically, I suggest following this order (some additions/changes are also indicated for each subparagraph).

  In response to this suggestion, we have restructured the description of our methods as described below.

2.1 SOA generation – In addition to describe the aerosol generation experiment, as they already do, the authors should also provide and briefly discuss (in the supplement, not in the main text) data related to O3, NOx, AMS, particle size, as it will help other researchers reproducing their results. Furthermore, the chamber used in their experiment should be described (or a reference should be provided to a published description of their chamber). The *online* AMS analysis (now part of section 2.3.2) should also be in this section (i.e., in the supplement), not as part of the main text (unless the authors wish to discuss AMS data as a part of their results; in this case, a comparison with HRMS and UV-vis data of filters pre- irradiation should also be included). If all this data was already presented and discussed in a publication, they can simply add the reference to the published work. In addition, description of how blank samples are obtained should be added as well.

  All data for ozone, NOx, SMPS, and online-AMS collected during our chamber experiments have now been made available on the Index of Chamber Atmospheric Research in the United States (ICARUS), and we thank the reviewer for reminding us to cite the data set. Additionally, a reference describing the chamber was added. Blanks in this work were simply filters without SOA collected onto them, so we chose not to include their description in the SOA generation section. Instead, we have included their description in the HRMS section on line 216.

2.2 Photolysis experiments – Here, the authors should describe the general features of the irradiation setup used for both aqueous and on-filter photolysis (i.e., xenon lamp, now part of section 1.2.1). In addition to the information already provided, the authors should also add a description of their irradiation experiments – How long where these experiments run for? When did the authors collect aliquots for which analyses? How many replicates were performed?

  This information has been moved to this section. Additionally, the number of replicates and a reference to Table 1 were added. We did not add a description of aliquot collection because the whole sample was analyzed for each analysis. We have clarified this is how the photolysis experiments were performed in lines 177-179.

2.2.1 Aqueous experiments – Here, the authors should describe specific details related to the aqueous-phase experiments, including pre- and post-irradiation sample preparation (now, also in section 1.2.1)

  This section has been revised accordingly.

2.2.2 On-filter experiments – Here, the authors should add details specific to irradiation of filters, including post-irradiation sample preparation for further analyses (now part of section 2.3.1)

This section has been revised accordingly.

2.3 Sample and data analyses
2.3.1 UV-vis analyses – Here, there should be a description of the UV-vis spectrometer and data analysis protocol. In the current form, this section requires some changes - in my opinion, there are too few details on actual measurements and too many details on recovery calculation. Missing information on sample collection include the following: how was the UV- vis operated (single beam or double beam)? If in double-beam, which reference was used? Which cuvettes were used (1-cm quartz)? How many replicates were collected for each datapoint? Missing information on data analyses include the following: Did you perform blank subtraction? (If yes, which blank spectrum did you use?) Did you correct for long-wavelength absorption? Did you correct for changes occurring in the dark? (According to Figure S3, there are some substantial variations in UV-vis absorption in the ammonium sulfate kept in the dark.) What is the range of absorbance you detected at your experimental concentrations? (i.e., was within the linear range of the instrument?) Furthermore, the text on estimating SOA recovery from UV-vis analyses should be moved to the supplement.

We have added in the information requested here, included the cuvette dimensions and description of reference solutions. We did not discuss changes occurring in the dark here because we report the unadjusted spectra in Figures 1a, S3, and S4. We have added a description of correcting for changes in the dark in the discussion of the UV-Vis results in lines 263-265.

2.3.2 UPLC-PDA-HRMS analyses – Here, add content of section 2.2.3. Additional data analyses performed (see previous comments) should also be described here.

This section has been revised, and we have added information about the blank preparation and updated the PDA analysis description to match our new PDA analysis.

2.3.3 FT-IR analyses – Please, provide instrument setting, sample analyses, and data analyses details here. This information is currently missing.

This section has been moved from the SI to the main text as suggested and this information has been added.

2.3.4 Offline AMS analyses – Here the authors can refer to the O'Brien et al. publication describing the system they used.

This section has been revised and this reference is included.

Below, I provide a list of minor changes that the authors may wish to address in preparing their revised manuscript.

*Abstract and introduction*

**Lines 8-9:** Please, change "oxides of nitrogen" to "oxides of nitrogen ($NO_X$)".

We thank the reviewer for this suggestion. This has been done.

**Line 14**: Please, change "UV-B" to "UV" (based on your spectrum in Figure S1, you have also a substantial amount of UVA during your experiments).

This is true, the change has been incorporated.

**Line 24:** Please, change "nitrophenol" to "nitrophenols".

*We have changed the sentence to read "photodegradation of nitrophenols."*

**Line 33:** Please, change "NO$_x$" to "nitrogen oxides (NO$_x$)"

*We thank the reviewer for catching this. It has been fixed.*

**Line 39**: Please, specify what "non-cloud conditions" refer to.

*This has been updated to clarify RH < 100%*

**Line 44:** Please, change "aqueous particles" to "deliquesced aerosol particles".

*We thank the reviewer for this suggestion. This has been done.*

**Line 45**: Please, change "particle phase water" to "water present in deliquesced aerosol particles".

*We thank the reviewer for this suggestion. This has been done.*

**Line 62**: Please specify why ammonium sulfate is a more interesting/relevant salt to use compared to others already investigated in the literature.

*We thank the reviewer for this suggestion. This has been done in lines 61 to 62.*

**Lines 65-66**: The authors may want to point out that also the concentration range of the organics is different, not only the ionic strength; this can have effect on products formation (e.g., oligomers formation is more likely when solutions are more concentrated).

*While it is true that aerosols have higher concentrations of organics because of lower water content than in a cloud, we do not address this in our manuscript so we do not believe it will help our argument. We have added a clarification to the aqueous photolysis section that the organic concentrations used here are lower than the concentrations in particulate matter (line 143).*

**Line 67**: Please, specify if "near UV" refers to UVA, UVB, or both. Furthermore, in which phase are the SOAs in the discussed studies? (i.e., was the irradiation performed on filters or in the aqueous phase?)

*These clarifications have been added in lines 67 and 68.*

**Line 79**: Please change "This" to "This hypothesis".

*This change has been incorporated.*

*Materials and methods*

**Line 97:** How was hydroxyl radicals steady-state concentrations measured/estimated? This information should be added in the supplement (where additional information related to section 2.1 of the materials and methods are presented).

*A new section describing this has been added to the SI along with information about the PTR-MS used to do the analysis in lines 100 to 104.*

**Line 99 and 100:** Is "NO$_y$" a typo? Why not "NO$_x$"?

*This is not a typo. The instrument detects all oxidized nitrogen compounds (except for N$_2$O), including NO, NO$_2$, organic nitrates, nitrous acids, and nitric acid, collectively known as NO$_y$.*

**Table 1:** Please, change "location" to "location of further analyses or experiments" in the heading of the first column (or add a new column named "location of the experiment").

*This change has been made.*

**Lines 120-121:** Did the extracts dissolve easily in water (or ammonium sulfate solution), or did the authors sonicate the filters? (If the authors sonicated, for how long and at which power?). The authors should also clarify which type of water was used to prepare their aqueous extract (i.e., deionized water or milliQ; if MilliQ water, information on the generation system should also be provided).

*The SOA extracts dissolved easily so no shaking or sonification was necessary. MilliQ water was used. These clarifications have been added in lines 138 to 140.*

**Lines 121, 124, 142**: Please, change "µg/mL" to "mg/L". This applies also to instances in the supplement (e.g., Table S2).

*This change has been made.*

**Line 124**: I found it surprising that also the water solution has such a low pH (generally, the pH of deionized water is 5.5 – 6.5 due to dissolved CO2). Can you explain this lower pH? (e.g., is it due to organic acids parts of your aerosol or to inorganic components? Can the AMS data provide useful insights?)

*I believe it's from organic acids. 4.5 or 5 is often what we measure for aerosol dissolved in water at this concentration. Our solutions are something like 20 to 25% SOA, so there are likely high concentrations of organic acids.*

**Lines 133-134**: Please, clarify if the cuvette was irradiated from the top or from the side.

*It was from the side. This clarification has been made.*

**Line 135:** A typical approach to remove salts and prepare samples for HRMS analysis involve solid phase extraction (SPE). Have the authors tried/considered this procedure as well? Are there literature precedents showing that your procedure is better/more convenient?

*This is a good question. For aerosol samples, SPE for inorganic salt extraction has been shown to reduce the recovery of more highly oxygenated and polar compounds, which are important in SOA composition. We have added a discussion of this to lines 149 to 153. Note that our recovery from ammonium sulfate is also incomplete but we quantify it in the text.*

**Line 144:** Please, check that "g/mL" is the correct concentration unit.

*These are the correct units, but we have changed it to g cm$^{-3}$ to make the conversion to MAC in cm$^2$ g$^{-1}$ more apparent.*

**Line 166:** Please, provide the brand of the UPLC instrument.

*This addition has been made.*

**Lines 171 – 173:** I would add that separating the compounds by LC previous to HRMS allows one also to improve characterization – as you can assign a formula to each peak of the chromatogram.

*We thank the reviewer for the suggestion. This has been added.*

*Results, discussion, and conclusions*

**Figures 1-2**: I suggest combining Fig. 1 and Fig.2 in a single 2-panel figure, at it will make it easier for the reader to see that Fig. 2 was obtained using data from Fig. 1. Furthermore, I suggest changing the x-axis of Fig. 2 to "time (hours)" (instead of "(min)"), as this is the same unit in the legend of Fig. 1. In the caption related to Fig. 1 data, the authors should specify the conditions under which this data was acquired, instead of referring to "representative conditions" (i.e., say directly that it is was collected during an aqueous-phase experiment in water). In addition, a reference to spectra collected in other conditions (i.e., Figure 3) should be added to the caption.

Figures 1 and 2 have been combined as suggested and the units on the x-axis have been changed. The reference to Figure S4 has been added to line 257 and the description for Fig 1(a) has been shortened to state the spectra are from the photolysis in water.

**Line 264**: Please, provide empirical evidence for the following statement: "[…] the fastest absorbance decrease with photolysis was observed at 350 nm" – from analysis of their UV-vis data, the authors should be able to justify this choice.

With the reviewer's previous suggestions to perform our analyses on the integrated wavelength range from 300 to 700 nm, this sentence has been changed to exclude this phase.

**Figure 3**: I have two suggestions here: 1. Remove one panel (I found it confusing that the black and blue traces are shown twice); to increase readability, the authors may consider increasing the offset of each peak and normalize the PDA counts to one given, convenient, signal; 2. Indicate on the graphs which are the peaks for which you could assign a molecular formula (and add the formulas/names).

For suggestion 1, panel (a) and panel (b) represent two trials from two different filters, which we used to analyze the reproducibility of our analyses. We have clarified this goal in the figure caption. Additionally, we agree that normalization would be beneficial for the readability of the plot, but as there is no unphotolyzable reference in our samples, so we do not have a requisite signal for normalization. We have increased the offset of the peaks in panel (b) as suggested.

For suggestion 2, this is a very good idea. We have added vertical lines indicating the positions of the assignable peaks along with their assigned molecular formulas.

**Line 295:** Please, explain how you assessed the reproducibility of your analyses.

We have clarified that we are talking about the heights of the major peaks as well as the shape of the peak distribution in our samples after our extraction process in line 339.

**Line 313 – 315**: Here, a total organic carbon analysis before and after irradiation could provide some evidence about this point. If the authors did this analysis or have material left to perform this analysis, it would be good addition to their experimental dataset.

We agree with the reviewer on this point; this would be an excellent addition to our dataset. Unfortunately, due to the inefficiency of toluene's SOA production, we do not have any material left over. Additionally, we do not have a functioning TOC available to us for this analysis.

**Figures 4 and 5**: As for the two panels in Figure 3, I advise the authors to produce one single figure with all four experimental conditions.

We thank the reviewer for the suggestion. However, the spectra in Figures 3 and 4 represent two different experiments and therefore have slightly different peak distributions, making them inconvenient to directly compare. Additionally, we believe it is a valuable contribution to our paper to point out that the mass spectrometry results for both the unaged

and the photolyzed in water samples are very similar between the different experiments. We have better illustrated this in lines 366-367.

**Line 343 – 344:** Did the authors record FTIR data also for aqueous samples? Can their offline AMS method distinguish between inorganic (e.g., HNO3) and organic nitrogen-containing compounds? If not, inorganic nitrogen containing products may remain in the aqueous phase contributing to $NO^+ + NO_2$ counts.

The aqueous samples were too dilute to be analyzed using FTIR, so we do not have FTIR results for these samples. This explanation has been added to the SI section on the FTIR results. The offline AMS method is not able to directly distinguish inorganic and organic nitrogen, but inorganic and organic nitrogen groups fragment into different ratios of these two ion signals. We have added a discussion of this in lines 420 to 428 of the main text along with two new panels in Figure 6.

**Figure 6**: Please explain why the ammonium sulfate data was not included in this analysis.

The purpose was to contrast between filter and water. We did not observe differences in the nitrogen changes between water and ammonium sulfate. Additionally, this test was performed on just one filter split in half to give better control over the SOA composition. It was not possible to also include an ammonium sulfate condition with the same filter because there was not enough mass on the filter. We have added this explanation to lines 407-411.

**Section 3.2.4 and lines 393, 414-415**: Please, make clear that the ones discussed are only *proposed* mechanisms based on previously reported reactivity for nitrophenols. The authors did not perform any experiment to test the underlying mechanism. Instead, this point should be highlighted as an area for further research in the final section. I also highlight that running more analyses to confirm the identity of some of the well-resolved peaks (see my comments above) will allow the authors to collect a piece of evidence for the existence of triplet reactivity – if compounds that they identify are the same for which triplet reactivity has been *observed*, not only speculated, in previous work.

Thank you for the suggestion. We have added an explanation that we are only proposing plausible mechanisms based on previous work at the beginning of section 3.2.4 (lines 440-441) and changed the last sentence in the section to say that we propose that the triplet chemistry explains our observed results (line 475). In the conclusions section, we have clarified that we are proposing these mechanisms as an explanation for our results based on previous studies (lines 490 to 491) and added a sentence stating that further work needs to be done to confirm the proposed mechanistic explanation (lines 521-523).

**Lines 375 – 379 (and lines 414-415):** Photodegradation of dissolved organic matter generates carbonyls (e.g., Moran and Zepp[4] or Zhu and Kieber[5]). If the same happens during SOA to form new nitrogen-containing compounds. Did the authors consider this mechanism while explaining their ammonium sulfate results?

Indeed, reactions between ammonia/ammonium and carbonyl groups are well known to cause browning, and we do observe an increase in absorbance at 300 nm in the samples aged with ammonium sulfate in the dark. We have added a discussion of these processes in lines 466 to 475, as well as adding a brief statement to lines 513-514 in the conclusion.

**References**

(1)    Laszakovits, J. R.; MacKay, A. A. Data-Based Chemical Class Regions for Van

Krevelen Diagrams. *J. Am. Soc. Mass Spectrom.* **2022**, *33* (1), 198–202. https://doi.org/10.1021/jasms.1c00230.

(2)     Merder, J.; Freund, J. A.; Feudel, U.; Hansen, C. T.; Hawkes, J. A.; Jacob, B.; Klaproth, K.; Niggemann, J.; Noriega-Ortega, B. E.; Osterholz, H.; Rossel, P. E.; Seidel, M.; Singer, G.; Stubbins, A.; Waska, H.; Dittmar, T. ICBM-OCEAN: Processing Ultrahigh-Resolution Mass Spectrometry Data of Complex Molecular Mixtures. *Anal. Chem.* **2020**, *92* (10), 6832–6838. https://doi.org/10.1021/acs.analchem.9b05659.

(3)     Del Vecchio, R.; Blough, N. V. Photobleaching of Chromophoric Dissolved Organic Matter in Natural Waters: Kinetics and Modeling. *Mar. Chem.* **2002**, *78* (4), 231–253. https://doi.org/10.1016/S0304-4203(02)00036-1.

(4)     Moran, M. A.; Zepp, R. G. Role of Photoreactions in the Formation of Biologically Labile
Compounds from Dissolved Organic Matter. *Limnol. Oceanogr.* **1997**, *42* (6), 1307–1316. https://doi.org/10.4319/lo.1997.42.6.1307.

(5)     Zhu, Y.; Kieber, D. J. Wavelength- and Temperature-Dependent Apparent Quantum Yields for Photochemical Production of Carbonyl Compounds in the North Pacific Ocean. *Environ. Sci. Technol.* **2018**, *52* (4), 1929–1939. https://doi.org/10.1021/acs.est.7b05462.

---

## Referee Report (RR1)

I thank the authors for considering and implementing my suggestions, especially with respect with structure of the materials and methods, and interpretation of UV-vis and MS data.

Before accepting the manuscript for publication, I advise the authors to clarify a few minor points that I highlighted in orange in the comments list below. The remaining items mostly are typos and suggested rephrasing (for clarity).

**Abstract and intro**
**Line 23**: Faster photobleaching with respect to ..? Please, add the comparison term.
**Line 48-49:** Rephrase as following: "For example, photodegradation of pyruvic acid in high ionic strength and low pH values (< 4) results in the red shift of its major absorption band." (it is unclear what "peak intensity" is: is it increase in the molar extinction coefficient of its absorption band? Clarify or remove)
**Line 53:** Can you specify the wavelengths?
**Line 60:** Rephrase as following: "However, no photodegradation studies have been conducted in the presence …"
**Line 62-63:** Change "broken down" to "degraded"
**Line 72:** On which substrate was the photolysis conducted? Please, change "OH radicals" to "hydroxyl (OH) radicals." (OH has not yet been defined)
**Line 72:** Change "with strongly" to "concluding that strongly"
**Line 79:** Change "dry" to "dry (i.e., organic)"
**Line 88:** Change "slowed under these conditions" to "slowed as compared to the experiment in pure water."

**Materials and methods**
**Line 96:** Remove the comma after "chamber" (or change as following: "… chamber described in Malecha and Nazadorovick, 2017")
**Line 130:** Add "Further experimental details on the two setups are provided in the following sections."
**Line 139:** Move "water" right after "MilliQ"; How much water did you add/how much total volume did you obtain?
**Line 149:** Change "solvent" to "water"
**Lines 156-160:** From my experience, (pure) water and acetonitrile are quite soluble with each other's: how did you manage to separate the two solvents after adding the 2.5 mL of ACN? Was the presence of salt in water helping phase separation? If so, how could you reproduce the same procedure with the pure water isolate? I recommend clarifying these points in lines 156-157.
**Line 168:** Why was this filter extracted in ACN (while for the others you used water)? Do you think this may introduce a bias in your UV-vis spectra? Do you have data showing that UV-vis spectra in ACN and water are similar for these filter extracts?
**Line 174:** Change "with photolysis" to "during photolysis"
**Line 177:** Change "the whole setup" to "the cuvette"
**Line 180-182:** I think you can delete the part in parenthesis: it is sufficient to say that you used the same 0.5 cm cuvette. I would rather add that these samples were measured in ACN.

**Results and discussion**

**Figure 1b:** Can you specify in the caption if this graph was obtained using the dark-corrected or the uncorrected spectra?

**Line 271:** Change "absorbance" to "normalized absorbance"; change "rate constants" to "first-order rate constants"

**Line 272:** Change "absorbance" to "normalized absorbance

**Paragraph starting at line 275 (until "… from the data."):** Rephrase as following: "Based on this analysis, we observed a considerably slower photolysis on filter than in the aqueous phase. However, for the filter sample we could only collect 4 datapoints, which may introduce a bias in our analysis. In particular, the filter data did not include the 0.5 h timepoint, which characterizes the fast-reacting chromophores pool in the aqueous sample."

**Table 2:** I suggest reporting the results with 2 significant digits if the first significant digit of the error is < 4. In other words, I suggest adding an additional significant digit to all tau1 values and of tau2 of AS. Furthermore, I suggest moving the lifetime of the filter under tau_2, as, based on the text above, you are only obtaining the "slow reacting" pool. (you can consider changing "A_2" to "A_1" in line 277.

**Line 301:** Change "A large" to "Under all conditions, a large"

**Line 306:** Why for some peaks you give a formula but not a name? Is it correct that there are two C7H7NO3 compounds in Figure 2 (but only the one at 10.36 is identified as nitrocresol)? Can you indicate with an asterisk the formulas for which you could identify the associated chemical name in Figure 2?

**Figure 2:** I now understand the rationale for showing twice the unaged and water photolysis results. However, I am surprised to see such a large discrepancy from the two filters for the same treatment (e.g., unaged). (In panel (a), all peaks also appear less resolved, maybe due to some chromatographic issues, which makes them appear more different than they probably are). Can you add a sentence to the main text (maybe at the end of section 3.2.1) explaining why the unaged chromatograms of the two filters are different? Will this affect your conclusions?

**Line 336:** Change "on filter" to "on filter (from filter 3)"

**Line 337:** Change "sulfate" To "sulfate (from filer 4)"

**Line 344:** I suggest pointing out that these are also the compounds identified in Figure 2 as being the most abundant (provide the names as well, if possible)

**Line 345 and entire section 3.2.2:** How did you evaluate the difference between the various conditions? Is a visual/qualitative comparison of the highest peak intensities? The same comment applies also for the discussion of VK diagram results.

**Figure 3 and 4, caption:** Clarify that it is from Filter 3 and 4, respectively

**Figure 5:** I appreciate that the authors took up my suggestion of using KV diagrams. I have a follow up suggestion: for the figure in the main text, do you think it be useful to show only datapoints above a certain intensity threshold? This will allow cleaning up the diagrams and better highlight differences among treatments.

**Line 406:** Change "with on-filter" to "during on-filter"

**Line 407:** Change "The purpose of this analysis was to directly contrast the difference in composition changes with photolysis" to "Differently from HRMS, this analysis allowed characterizing bulk changes in composition during photolysis"

**Line 408:** Change "observe dramatic differences" to "dramatic changes in chemical composition"

**Line 431:** Then you think that during on-filter photolysis nitro groups are converted to gas-phase N species? It would be good to add some sort of conclusion to the discussion.

**Line 458-459:** Please revise as following: "In spite of the *slower* nitrophenols photodegradation (assessed via HRMS, offline-AMS, and FTIR analyses), the overall photobleaching (assessed via UV-vis analyses) was actually *faster* in the aqueous phase as compared to on-filter photodegradation."

**Line 463:** This fact may also agree with the larger decrease in the unresolved baseline absorption that you observed for the aqueous samples (- 35% and – 45% for pure water and AS) as compared to the filter isolates (- 21%). It would be good to point this out.

**Line 483:** Change "and a large fraction of photorecalcitrant fraction is likely." To "and likely results in the formation of a photorecalcitrant faction."

**Line 484:** Change "by SOA." To "of this type of SOA."

**Line 488:** Change "faster photobleaching" to "faster overall photobleaching (assessed by UV-vis analysis)."

**Line 490**: Change as following: "Based on previous studies, we propose that the difference … sample matrices."

**Line 500:** remove "photochemical"

**Line 505-504:** "(Hems et al., 2021)" should be placed after "days".

**Line 502:** Change "photobleaching" to "overall photobleaching"

**Line 504:** delete "with respect to photobleaching and lifetimes"

---

## Author Response (AR2)

(accepted as is)

The authors considered the reviewers comments and substantially improved the manuscript. Therefore, I suggest publication.

Thank you for recommendation. We did make additional changes in response to the suggestions by the anonymous referee #2 (described in a separate document).

**Anonymous referee #2**

(accepted subject to minor revisions)

I thank the authors for considering and implementing my suggestions, especially with respect with structure of the materials and methods, and interpretation of UV-vis and MS data.

Before accepting the manuscript for publication, I advise the authors to clarify a few minor points that I highlighted in orange in the comments list below. The remaining items mostly are typos and suggested rephrasing (for clarity).

**Abstract and intro**
**Line 23**: Faster photobleaching with respect to ..? Please, add the comparison term.
This change has been made.
**Line 48-49:** Rephrase as following: "For example, photodegradation of pyruvic acid in high ionic strength and low pH values (< 4) results in the red shift of its major absorption band." (it is unclear what "peak intensity" is: is it increase in the molar extinction coefficient of its absorption band? Clarify or remove)
This statement has been updated.
**Line 53:** Can you specify the wavelengths?
As the two studies looked at different compounds in the presence of different salts, they observed different wavelengths. We therefore feel it would muddle the point of the sentence to add specific wavelengths for each study.
**Line 60:** Rephrase as following: "However, no photodegradation studies have been conducted in the presence …"
Thank you for the suggestion. It has been implemented.
**Line 62-63:** Change "broken down" to "degraded"
This change has been made on line 67, which we believe is what the reviewer is referring to.
**Line 72:** On which substrate was the photolysis conducted? Please, change "OH radicals" to "hydroxyl (OH) radicals." (OH has not yet been defined)
Thank you for catching this. It has been fixed.
**Line 72:** Change "with strongly" to "concluding that strongly"
This has been added.
**Line 79:** Change "dry" to "dry (i.e., organic)"
This clarification has been made.
**Line 88:** Change "slowed under these conditions" to "slowed as compared to the experiment in pure water."
This adjustment has been made.

Materials and methods
**Line 96:** Remove the comma after "chamber" (or change as following: "… chamber described in Malecha and Nazadorovick, 2017")

This has been done.

**Line 130:** Add "Further experimental details on the two setups are provided in the following sections."

Thank you for the suggestion. We have included it.

**Line 139:** Move "water" right after "MilliQ"; How much water did you add/how much total volume did you obtain?

This has been updated.

**Line 149:** Change "solvent" to "water"

This change has been made in line 154, which we believe is what the reviewer is referring to.

**Lines 156-160:** From my experience, (pure) water and acetonitrile are quite soluble with each other's: how did you manage to separate the two solvents after adding the 2.5 mL of ACN? Was the presence of salt in water helping phase separation? If so, how could you reproduce the same procedure with the pure water isolate? I recommend clarifying these points in lines 156-157.

We are sorry for the confusion. There was no phase separation into aqueous and organic fractions. Rather, there was very little water present after rotary evaporation (maybe a few microliters), and when ACN was added most of the ammonium sulfate precipitated (because it is not as soluble in ACN) and the remaining water was removed with the ACN. This has been clarified in the revised text.

**Line 168:** Why was this filter extracted in ACN (while for the others you used water)? Do you think this may introduce a bias in your UV-vis spectra? Do you have data showing that UV- vis spectra in ACN and water are similar for these filter extracts?

We are sorry for being unclear. Both sets of filters were extracted in ACN. However, in the case of the aqueous samples, the ACN was removed and water or 1 M ammonium sulfate was added to the sample before photolysis. We have added a clarification to the section about the aqueous photolysis experiments restating that the filters were extracted using ACN. All UV-Vis data for the filter experiments reported here was taken in ACN and the comparison between the water and ACN solvents can be seen in Fig. S4. They are very similar.

**Line 174:** Change "with photolysis" to "during photolysis"

This has been changed.

**Line 177:** Change "the whole setup" to "the cuvette"

This has been changed.

**Line 180-182:** I think you can delete the part in parenthesis: it is sufficient to say that you used the same 0.5 cm cuvette. I would rather add that these samples were measured in ACN.

This has been deleted and we have added in that the UV-Vis measurements were conducted in ACN for the filter samples.

**Results and discussion**

**Figure 1b:** Can you specify in the caption if this graph was obtained using the dark-corrected or the uncorrected spectra?

Thank you for the suggestion. We have updated the caption.

**Line 271:** Change "absorbance" to "normalized absorbance"; change "rate constants" to "first-order rate constants"

Thank you for the corrections. We have implemented them.

**Line 272:** Change "absorbance" to "normalized absorbance

Thank you for the correction. We have implemented it.

**Paragraph starting at line 275 (until "… from the data."):** Rephrase as following: "Based on this analysis, we observed a considerably slower photolysis on filter than in the aqueous phase. However, for the filter sample we could only collect 4 datapoints, which may introduce a bias in our analysis. In particular, the filter data did not include the 0.5 h timepoint, which characterizes the fast-reacting chromophores pool in the aqueous sample."

We have rephrased as suggested.

**Table 2:** I suggest reporting the results with 2 significant digits if the first significant digit of the error is < 4. In other words, I suggest adding an additional significant digit to all tau1 values and of tau2 of AS. Furthermore, I suggest moving the lifetime of the filter under tau_2, as, based on the text above, you are only obtaining the "slow reacting" pool. (you can consider changing "A_2" to "A_1" in line 277.

Thank you for the recommendations and noticing an inconsistency in indexing the coefficients. They have been added.

**Line 301:** Change "A large" to "Under all conditions, a large"

Thank you for the suggestion. This has been completed.

**Line 306:** Why for some peaks you give a formula but not a name? Is it correct that there are two C7H7NO3 compounds in Figure 2 (but only the one at 10.36 is identified as nitrocresol)? Can you indicate with an asterisk the formulas for which you could identify the associated chemical name in Figure 2?

Only the 5 most intense peaks were originally given a name. We have updated the paragraph so all peaks for which the name can be reasonably assigned based on previous studies are given a name. The two peaks which are not named are marked with an asterisk.

**Figure 2:** I now understand the rationale for showing twice the unaged and water photolysis results. However, I am surprised to see such a large discrepancy from the two filters for the same treatment (e.g., unaged). (In panel (a), all peaks also appear less resolved, maybe due to some chromatographic issues, which makes them appear more different than they probably are). Can you add a sentence to the main text (maybe at the end of section 3.2.1) explaining why the unaged chromatograms of the two filters are different? Will this affect your conclusions?

Over the course of these experiments, we experienced some degradation of the HPLC column, which is under very frequent use in this shared instrument. The samples in panel b were taken about a year before the samples in panel a. We believe we may have been able to identify and discuss more peaks if the column degradation had not occurred, but our conclusions as presented here will still be valid because we only included peaks which were reproducible between the two trials. We have excluded the data for peaks which we could not resolve in panel a. We have added a brief explanation of this to the paragraph preceding the figure.

**Line 336:** Change "on filter" to "on filter (from filter 3)"

This change has been implemented.

**Line 337:** Change "sulfate" To "sulfate (from filer 4)"

This change has been implemented.

**Line 344:** I suggest pointing out that these are also the compounds identified in Figure 2 as being the most abundant (provide the names as well, if possible)

Thank you for the suggestion. This has been added.

**Line 345 and entire section 3.2.2:** How did you evaluate the difference between the various conditions? Is a visual/qualitative comparison of the highest peak intensities? The same comment applies also for the discussion of VK diagram results.

The analysis was a qualitative comparison of visual observations. We have added this clarification in both places.

**Figure 3 and 4, caption:** Clarify that it is from Filter 3 and 4, respectively

Thank you for suggesting this clarification. We have implemented it.

**Figure 5:** I appreciate that the authors took up my suggestion of using KV diagrams. I have a follow up suggestion: for the figure in the main text, do you think it be useful to show only datapoints above a certain intensity threshold? This will allow cleaning up the diagrams and better highlight differences among treatments.

Thank you for the suggestion. We have removed points with intensities less than 1% of the maximum intensity in the unaged samples from the plot in the main text while keeping all points in the SI Van Krevelen diagrams.

**Line 406:** Change "with on-filter" to "during on-filter"

This has been changed.

**Line 407:** Change "The purpose of this analysis was to directly contrast the difference in composition changes with photolysis" to "Differently from HRMS, this analysis allowed characterizing bulk changes in composition during photolysis"

We have elected to keep the previous sentence.

**Line 408:** Change "observe dramatic differences" to "dramatic changes in chemical composition"

This has been updated.

**Line 431:** Then you think that during on-filter photolysis nitro groups are converted to gas-phase N species? It would be good to add some sort of conclusion to the discussion.

This is what we suspect, and we agree that this paragraph would benefit from a conclusion. We have added a discussion stating we think it is likely the N-containing products are being lost to the gas phase, and added a few relevant references about HONO formation from nitrophenols.

**Line 458-459:** Please revise as following: "In spite of the *slower* nitrophenols photodegradation (assessed via HRMS, offline-AMS, and FTIR analyses), the overall photobleaching (assessed via UV-vis analyses) was actually *faster* in the aqueous phase as compared to on-filter photodegradation."

Thank you for the suggestion. It has been implemented.

**Line 463:** This fact may also agree with the larger decrease in the unresolved baseline absorption that you observed for the aqueous samples (- 35% and – 45% for pure water and AS) as compared to the filter isolates (- 21%). It would be good to point this out.

Thank you for the suggestion. We have added a brief statement to this effect.

**Line 483:** Change "and a large fraction of photorecalcitrant fraction is likely." To "and likely results in the formation of a photorecalcitrant faction."

This has been changed.

**Line 484:** Change "by SOA." To "of this type of SOA."

Thank you for noticing this. We have updated it.

**Line 488:** Change "faster photobleaching" to "faster overall photobleaching (assessed by

UV-vis analysis)."

This has been updated.

**Line 490**: Change as following: "Based on previous studies, we propose that the difference … sample matrices."

This change has been made.

**Line 500:** remove "photochemical"

This has been updated.

**Line 505-504:** "(Hems et al., 2021)" should be placed after "days".

This has been moved.

**Line 502:** Change "photobleaching" to "overall photobleaching"

Thank you for the suggestion. This has been changed.

**Line 504:** delete "with respect to photobleaching and lifetimes"

Thank you for catching this. It has been changed.